# AutoEP: LLMs-Driven Automation of Hyperparameter Evolution for Metaheuristic Algorithms

**Zhenxing Xu**[1,*] **Yizhe Zhang**[1,*] **Weidong Bao**[1,†] **Hao Wang**[1]**, Ming Chen**[2]**,**
**Haoran Ye**[3]**, Wenzheng Jiang**[1]**, Hui Yan**[4]**, Ji Wang**[1]
[1]National Key Laboratory of Big Data and Decision, National University of Defense Technology
[2]Department of Automation, Tsinghua University
[3]School of Intelligence Science and Technology, Peking University
[4]Information Support Force Engineering University
{xuzhenxing,zyz,wdbao,whao199,jiangwenzheng,yanhui13,wangji}@nudt.edu.cn
cmself@163.com, hrye@stu.pku.edu.cn

## Abstract

Dynamically configuring algorithm hyperparameters is a fundamental challenge in computational intelligence. While learning-based methods offer automation, they suffer from prohibitive sample complexity and poor generalization. We introduce AutoEP, a novel framework that bypasses training entirely by leveraging Large Language Models (LLMs) as zero-shot reasoning engines for algorithm control. AutoEP's core innovation lies in a tight synergy between two components: (1) an online Exploratory Landscape Analysis (ELA) module that provides real-time, quantitative feedback on the search dynamics, and (2) a multi-LLM reasoning chain that interprets this feedback to generate adaptive hyperparameter strategies. This approach grounds high-level reasoning in empirical data, mitigating hallucination. Evaluated on three distinct metaheuristics across diverse combinatorial optimization benchmarks, AutoEP consistently outperforms state-of-the-art tuners, including neural evolution and other LLM-based methods. Notably, our framework enables open-source models like Qwen3-30B to match the performance of GPT-4, demonstrating a powerful and accessible new paradigm for automated hyperparameter design. Our code is available at https://github.com/YiZheZhang12/AutoEP.

## 1 Introduction

The performance of complex algorithms, from numerical optimizers to machine learning models, is critically governed by their internal hyperparameters. Dynamically adapting these parameters to suit the problem instance at hand represents a long-standing challenge in automated algorithm design Wu et al. (2024; 2025). Metaheuristic algorithms, a cornerstone of solving complex combinatorial problems, serve as a canonical example. Their effectiveness hinges on a delicate balance between exploration (diversifying the search) and exploitation (intensifying the search in promising regions), a trade-off directly controlled by their hyperparameter configurations Eiben et al. (2002). Mastering this balance is crucial for achieving state-of-the-art performance but remains a formidable open problem.

Traditional approaches to dynamic hyperparameter adaptation fall into two categories. Manual, rule-based strategies Eiben et al. (2002); Thierens (2005); Ansótegui et al. (2009); Joshi & Bansal (2020); Leon & Xiong (2015); Li et al. (2025); Shao et al. (2025) embed human expertise into hard-coded logic, but are brittle, labor-intensive, and fail to generalize across different problems or algorithms. To overcome this, data-driven methods, particularly deep reinforcement learning (DRL) Tessari & Iacca (2022); Ma et al. (2024); Liu et al. (2023); Tatsis & Parsopoulos (2020); Yin et al. (2021), have attempted to learn adaptive policies from scratch. However, this paradigm faces fundamental

---

*Equal contribution.
†Corresponding author.

limitations: (1) prohibitive sample complexity, requiring millions of algorithm executions to train a single policy, and (2) poor generalization, where policies often overfit to the training distribution of problems and fail on unseen instances or algorithm variants Guo et al. (2024). This reveals a critical gap: the need for a framework that can adapt algorithm behavior without requiring expensive, instance-specific training.

The recent advent of Large Language Models (LLMs) offers a paradigm shift. Unlike traditional learning models that learn policies from scratch, LLMs distill vast amounts of knowledge from pre-training on extensive corpora of text and code. This process endows them with powerful emergent reasoning capabilities and a rich prior understanding of abstract concepts like "convergence," "diversity," and "optimization" Liu et al. (2024); YE et al. (2024); Liu et al. (2025); Hu et al. (2025). We hypothesize that this pre-trained knowledge can be harnessed to reason about optimal hyperparameter adjustments in a zero-shot manner, bypassing the costly training phase that plagues DRL-based approaches. This transforms the problem from one of learning a control policy to one of prompting a reasoning engine.

In this paper, we introduce AutoEP, a novel framework that operationalizes this vision. AutoEP synergizes the quantitative analysis of Exploratory Landscape Analysis (ELA) with the qualitative reasoning of LLMs, creating a new paradigm for zero-shot hyperparameter configuration. It overcomes the limitations of both manual design and data-intensive learning models. Our key contributions are:

**(1) A Zero-Shot Paradigm for Algorithm Control:** We propose a novel, training-free framework where LLMs act as a "pluggable" reasoning core to dynamically control algorithm hyperparameters. This general-purpose approach is applicable to any metaheuristic algorithm without modification or costly retraining (see Figure 1 for an overview).

**(2) Grounding LLM Inference with Search Trajectory Analysis:** To mitigate hallucinations and ensure data-driven decisions, we ground the LLM's inference in empirical evidence from the real-time search trajectory. We achieve this by continuously supplying the LLM with quantitative metrics, such as ELA features and historical decision data. These metrics, including fitness distribution, solution diversity, and search difficulty estimates, provide the LLM with a concrete awareness of the current optimization state. This process anchors the model's abstract reasoning in the observable dynamics of the search.

**(3) Complex Reasoning via Collaborative Open-Source LLMs:** We demonstrate that a collaborative pipeline of smaller, locally-deployed open-source LLMs (e.g., Qwen-72B, DeepSeek-67B) can effectively decompose and solve complex control tasks. Our empirical results show that this approach achieves performance comparable to that of large-scale proprietary models, such as GPT-4, while exhibiting significantly lower inference latency for the hyperparameter tuning task. This design substantially enhances the accessibility, reproducibility, and overall efficiency of sophisticated AI reasoning systems.

**(4) State-of-the-Art Performance Across Diverse Benchmarks:** Through extensive experiments on three distinct metaheuristics (GA, PSO, ACO Holland (1992); Dorigo et al. (2007); Kennedy & Eberhart (1995)) and four combinatorial optimization problems, we show that AutoEP consistently and significantly outperforms both traditional hyperparameter tuning methods and recent LLM-based approaches.

## 2 RELATED WORK

**Rule-Based and Heuristic Control.** Early attempts to automate hyperparameter tuning relied on hard-coded, rule-based heuristics. These methods embed expert knowledge into predefined rules that adjust parameters based on simple metrics like iteration count or population diversity Thierens (2005). For instance, strategies might deterministically increase mutation rates to escape local optima or adapt selection pressure over time Joshi & Bansal (2020). While an improvement over static settings, these approaches are fundamentally brittle. The underlying heuristics are problem-dependent and require

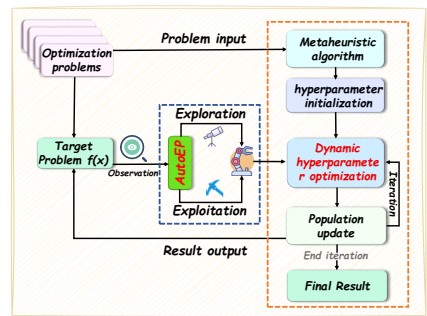

Figure 1: Process of hyperparameter tuning in metaheuristic algorithms using AutoEP.

extensive manual calibration. They lack the ability to adapt to unforeseen dynamics in the search process, making them unable to generalize across different problem classes or algorithms.

**Meta-Black-Box Optimization (Meta-BBO).** To address the inherent rigidity of manual heuristics, the field has shifted toward Meta-BBO Ma et al. (2023); Tatsis & Parsopoulos (2020); Guo et al.; Ma et al. (b); Li et al.. This paradigm integrates data-driven and reinforcement learning (RL) approaches to automate algorithm configuration and design. Unlike traditional optimization that searches within a decision space, Meta-BBO explores the algorithm space to discover optimal optimizers, typically leveraging historical data to learn adaptive strategies. While early frameworks like GLEET Ma et al. (2024) and NeuroCrossover Liu et al. (2023) successfully employed Deep RL for operator selection, they were constrained by high sample complexity. Recent advancements have expanded this domain; for instance, Neural ELA Ma et al. (2025) incorporates landscape features via neural exploratory landscape analysis, while DesignX Guo et al. (2025) proposes a dual-agent RL framework to jointly learn algorithm structure and hyperparameters. Despite these advances, current Meta-BBO methodologies remain heavily reliant on computationally expensive meta-training. This highlights a critical gap: the need for a zero-shot framework that retains the adaptivity of Meta-BBO while circumventing prohibitive training costs.

**LLMs for Algorithm Design: From Offline Generation to Online Control.** Leveraging pre-trained code knowledge, LLMs have advanced algorithm design significantly Ma et al. (a). Offline frameworks (e.g., EoH Liu et al. (2024), ReEvo YE et al. (2024)) successfully automate the generation of operators and configurations. However, optimization is inherently dynamic Surina et al. (2025), necessitating online adaptation. Recent attempts like EvoLLM Lange et al. (2024) employ LLMs as evolutionary operators for direct solution generation. Despite their novelty, such methods struggle with floating-point representation and context window constraints in high-dimensional search spaces.

This highlights a critical research gap. We argue that the optimal role for an LLM is not to mimic the solver, but to act as a high-level supervisor. Unlike prior works that replace search operators, we propose a framework for dynamic hyperparameter control. This design grounds the LLM's semantic reasoning in real-time search dynamics without being hindered by numerical precision issues. By doing so, we overcome the brittleness of rule-based heuristics and the sample inefficiency of RL, offering a robust and scalable solution for online control.

## 3 AUTOEP

### 3.1 GROUNDING REASONING WITH QUANTITATIVE SEARCH DYNAMICS

To effectively control a metaheuristic algorithm, the decision-making agent requires a real-time, quantitative understanding of the search process. As metaheuristics are black-box methods, we employ Exploratory Landscape Analysis (ELA) Mersmann et al. (2011); Ma et al. (2025) to extract features that characterize the algorithm's state. We selected a concise yet comprehensive set of features designed to capture four key aspects of the search: (1) the statistical distribution of the current population's fitness, (2) the structural properties of the local fitness landscape, (3) the diversity of solutions, and (4) the recent progress of the search.

#### 3.1.1 FITNESS DISTRIBUTION FEATURES.

**Skewness** ($S$). Measures the asymmetry of the solution distribution within the current population. It is calculated as follows:

$$S = \frac{\frac{1}{n} \sum_{i=1}^{n} (y_i - \bar{y})^3}{\left( \sqrt{\frac{1}{n} \sum_{i=1}^{n} (y_i - \bar{y})^2} \right)^3}, \tag{1}$$

where $y_i$ represent the fitness value of the $i$-th individual in the population, and $\bar{y}$ denote the mean fitness value of the population, where $n$ is the population size. For a minimization goal, a value near 0 suggests a balanced population. Positive skew ($S$>0) implies a long tail of low-quality solutions, indicating the search should intensify exploitation around the few discovered elites. Negative skew ($S$<0) suggests the population is converging on high-quality solutions and may be at risk of premature convergence, necessitating more exploration.

**Kurtosis** ($K$). Quantifies the "tailedness" of the solution distribution within the current population. It is calculated as follows:

$$K = \frac{\frac{1}{n}\sum_{i=1}^{n}(y_i - \bar{y})^4}{\left(\sqrt{\frac{1}{n}\sum_{i=1}^{n}(y_i - \bar{y})^2}\right)^4} - 3, \tag{2}$$

high kurtosis ($K{>}0$) indicates fitness values are tightly clustered with heavy tails, signaling low diversity and the need for exploration. Low kurtosis ($K{<}0$) implies a flat, dispersed distribution, suggesting exploitation is warranted to refine existing solutions.

### 3.1.2 FITNESS LANDSCAPE AND DIVERSITY FEATURES

**Meta-Model: Coefficient of Determination** ($R^2$). Quantifies the goodness-of-fit of a simple model (e.g., quadratic) to a sample of solutions, thereby assessing the structural predictability of the fitness landscape. It is calculated as:

$$R^2 = 1 - \frac{\sum_{i=1}^{n}(y_i - f(\vec{x}_i))^2}{\sum_{i=1}^{n}(y_i - \bar{y})^2}, \tag{3}$$

where $y_i$ is the fitness of the $i$-th individual, $\bar{y}$ is the mean fitness, and $f(x_i)$ is the fitness predicted by the model. A high $R^2 \approx 1$ indicates a well-structured landscape (i.e., a funnel), signaling a need to increase exploitation. Conversely, a low $R^2 \approx 0$ implies a rugged or multi-modal landscape, requiring an increase in exploration to avoid premature convergence.

**Dispersion Ratio** ($D_{ratio}$). Measures the population's diversity by comparing the spatial distribution of the best solutions against that of the worst solutions. A low ratio indicates convergence into a single promising region. It is calculated as follows:

$$D_{ratio} = \frac{D(Q_{\text{best}})}{D(Q_{\text{worst}})}, \tag{4}$$

where $Q_{\text{best}}$ and $Q_{\text{worst}}$ represent the sets of the best and worst solutions in the population based on a fitness quantile (top and bottom 10%), respectively. The function $D(Q) = \frac{2}{|Q|(|Q|-1)}\sum_{\vec{x}_i,\vec{x}_j \in Q, i<j} d(\vec{x}_i, \vec{x}_j)$ calculates the average pairwise distance among all individuals within a given set $Q$, using a distance metric $d(\cdot,\cdot)$ suitable for the decision space (e.g., Hamming distance for binary problems). A value of $D_{ratio} \ll 1$ (e.g., $< 0.2$) is a strong indicator of a single funnel structure, as the best solutions are tightly clustered. This signals the need to increase exploitation to refine the search within this promising basin. Conversely, a value of $D_{ratio} \approx 1$ suggests a multi-modal landscape, where elite solutions are found in disparate regions. This necessitates an increase in exploration to avoid premature convergence to a local optimum.

### 3.1.3 SEARCH PROGRESS FEATURE

**Variability** ($V$). The four indicators above describe the solution structure within the population and reflect the current state of the algorithm's optimization process. To capture the dynamics of the search more effectively, we design a rate-of-change indicator:

$$V = \frac{\frac{1}{m}\sum_{m=g-m}^{g-1}\bar{y}_m}{\bar{y}_g}, \tag{5}$$

to measure the evolutionary progress of the population, we introduce a rate of change indicator, $V$. Let $\bar{y}_g$ be the mean fitness of the population at generation $g$. The indicator $V$ quantifies the improvement in $\bar{y}_g$ relative to the mean fitness over the previous $m$ generations. For minimization problems, a value of $V > 1$ signifies sufficient progress, prompting the algorithm to intensify exploitation through local search. Conversely, $V \leq 1$ suggests that the population is stagnating, which triggers an increase in exploration to diversify the search.

These ELA features transform the black-box state of the metaheuristic into a structured, machine-readable format. This representation serves as the empirical foundation for the LLM's reasoning process, enabling it to make informed, data-driven decisions about hyperparameter adjustments.

## 3.2    A Closed-Loop Architecture for LLM-Driven Control

Figure 2: The AutoEP Framework.

AutoEP operates as a closed-loop control system that dynamically steers a metaheuristic algorithm. As shown in Figure 2, the framework iteratively performs three main functions: State-Sensing, Reasoning, and Action.

**1. State-Sensing and Context Formulation:** At each decision point, the framework captures the current state of the metaheuristic algorithm. This involves calculating the ELA features described in Sec 3.1. This real-time data is then combined with historical information from an Experience Pool. The pool stores a memory of past states, actions (hyperparameter settings), and their outcomes (fitness improvements). This collective information is formatted into a structured prompt that provides the LLM with both the current situation and relevant historical context.

**2. Reasoning via Multi-LLM Chain:** The formulated prompt is passed to our Chain of Reasoning (CoR) engine (detailed in Sec 3.3). This engine, composed of multiple collaborating LLMs, analyzes the current state in light of past experiences to determine whether to prioritize exploration or exploitation and translates this strategy into a concrete set of hyperparameter values.

**3. Action and Feedback:** The new hyperparameter configuration generated by the CoR engine is fed back to the metaheuristic algorithm, which uses it for the subsequent phase of the search. The performance outcome of this action is recorded and added to the Experience Pool, completing the feedback loop. This iterative process allows AutoEP to continuously adapt its strategy based on observed performance, effectively performing in-context learning throughout the optimization run.

## 3.3    Decomposing Control Logic with a Chain of Reasoning

Controlling a complex algorithm requires multi-faceted reasoning: understanding the task, diagnosing the current state, and deciding on a precise action. Entrusting this entire process to a single LLM with a monolithic prompt can lead to high inference latency and unstable outputs Wang et al. (2023). To address this, we introduce the CoR, a multi-LLM framework that decomposes the control task into a pipeline of specialized, more manageable reasoning steps. This approach not only improves performance through specialization Shen et al. (2023) but also enhances robustness through cross-validation. Our CoR pipeline consists of three distinct agents:

**The Strategist LLM (One-time Setup):** At the start of a run, the Strategist receives the problem description and the chosen metaheuristic algorithm, such as a GA. It generates a static "control mapping" that defines the qualitative effect of each hyperparameter (such as mutation rate and crossover probability) on the search process. For example, it might map the mutation rate to the concept of "boosting exploration". This map is generated once and serves as a foundational reference for the other agents.

**The Analyst LLM (State Diagnosis):** At each decision point, the Analyst LLM uses real-time ELA features and historical data from the Experience Pool to diagnose the current search state. It addresses the core question of whether to prioritize exploration or exploitation. To do this, it synthesizes ELA

signals to identify consensus (where multiple indicators, like low diversity and stagnation, both suggest exploration) or conflict (where indicators are contradictory, such as low diversity but rapid progress). Based on this diagnosis, it outputs a clear strategic directive, for instance, ACTION: Increase Exploration.

**The Actuator LLM (Decision and Tuning):** The Actuator receives the strategic directive from the Analyst (Increase Exploration) and the static control map from the Strategist. Its task is to translate the qualitative directive into a quantitative hyperparameter configuration. It performs this in two stages:

**Parameter Selection:** Using the control map, it identifies which hyperparameters to modify (e.g., increase mutation rate, decrease crossover probability).

**Magnitude Determination:** It determines the degree of adjustment. This is achieved through in-context learning, where it examines examples from the Experience Pool to infer effective tuning magnitudes from similar past situations. For instance, it might learn that small, incremental changes are better during stable progress, while large, aggressive changes are needed to escape deep stagnation.

This decomposed CoR pipeline transforms a complex, unstructured control problem into a series of focused, interconnected reasoning tasks, enabling more reliable and efficient automated algorithm configuration, as shown in Figure 3. The detailed prompts for each agent are provided in Appendix C.

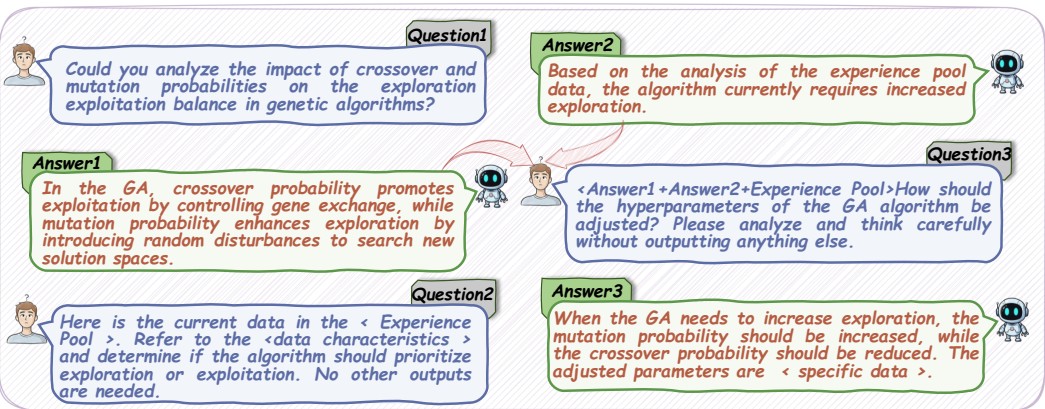

Figure 3: Demonstration of CoR.

## 4 EXPERIMENT

### 4.1 EXPERIMENTAL SETTINGS

To comprehensively evaluate AutoEP's performance, we selected classic combinatorial optimization datasets, including TSP Matai et al. (2010), CVRP Dantzig & Ramser (1959), and FSSP Emmons & Vairaktarakis (2012), as well as a more complex, realistic optimization task involving UAV-enabled IoT data collection Zhan et al. (2017). We compared AutoEP against three categories of algorithms:

**Hyperparameter tuning methods for metaheuristic algorithms.** PT Joshi & Bansal (2020) is a recent manually designed method for hyperparameter tuning. GLEET Ma et al. (2024) represents the SOTA in reinforcement learning-based hyperparameter tuning; we retrained the network strictly following the original experimental settings for different algorithms and datasets. BEA Lan et al. (2022) is a leading method using bayesian optimization for hyperparameter tuning.

**Neural combinatorial optimization.** DACT Ma et al. (2021) and LEHD Luo et al. (2023) are advanced methods for solving combinatorial optimization problems.

**LLMs-enhanced metaheuristic methods.** ReEvo and EoH are SOTA methods leveraging LLMs to enhance metaheuristic operators.

Experimental settings for AutoEP and the comparison algorithms are detailed in Appendix D. For performance testing on these datasets, both EoH and ReEvo used the GPT-3.5-turbo model, as

mentioned in their respective papers, while AutoEP utilized the Qwen3-30B LLMs. To ensure statistical robustness and mitigate the effects of random variation, all experiments were repeated 30 times. The results presented in this paper are the mean values from these runs.

## 4.2 RESULTS

### 4.2.1 VALIDATION ON TSP

Table 1: Comparison with various baselines on TSP. Opt.gap represents the percentage gap between the average run result and the optimal solution for this dataset; a smaller value is better. Time is the average runtime (unit: minute).

| Method | eil51 | | Rd100 | | Kroa150 | | rd300 | | rat575 | | dsj1000 | |
|---|---|---|---|---|---|---|---|---|---|---|---|---|
| | Opt.gap(%)↓ | Time | Opt.gap(%)↓ | Time | Opt.gap(%)↓ | Time | Opt.gap(%)↓ | Time | Opt.gap(%)↓ | Time | Opt.gap(%)↓ | Time |
| DACT | 0.00 | 0.6(m) | 0.09 | 4.1(m) | 0.13 | 7.9(m) | 0.93 | 18.7(m) | 2.55 | 26.3(m) | 4.97 | 71.5(m) |
| LEHD | 0.08 | 0.2(m) | 0.21 | 0.2(m) | 0.96 | 0.3(m) | 1.38 | 0.4(m) | 2.64 | 0.6(m) | 5.54 | 1.8(m) |
| GA | 1.47 | 0.6(m) | 3.61 | 0.9(m) | 5.26 | 1.7(m) | 11.33 | 2.8(m) | 14.75 | 3.3(m) | 21.94 | 5.3(m) |
| GA+PT | 0.33 | 0.7(m) | 1.61 | 0.9(m) | 3.94 | 1.7(m) | 8.82 | 2.8(m) | 9.43 | 3.3(m) | 19.25 | 5.3(m) |
| GA+GLEET | 0.07 | 1.2(m) | 1.49 | 1.5(m) | 3.23 | 2.4(m) | 7.11 | 3.7(m) | 8.06 | 4.9(m) | 16.23 | 6.8(m) |
| GA+BEA | 0.14 | 0.8(m) | 2.55 | 1.1(m) | 3.76 | 1.8(m) | 7.28 | 2.9(m) | 9.07 | 3.5(m) | 16.91 | 5.5(m) |
| GA+EoH | 0.31 | 0.6(m) | 1.38 | 1.1(m) | 3.61 | 1.9(m) | 7.16 | 3.1(m) | 8.32 | 3.4(m) | 19.39 | 5.3(m) |
| GA+ReEvo | 0.27 | 0.7(m) | 1.97 | 1.0(m) | 3.39 | 1.9(m) | 7.58 | 3.0(m) | 8.39 | 3.4(m) | 16.53 | 5.4(m) |
| **GA+AutoEP** | **0.11** | **3.1(m)** | **1.06** | **3.4(m)** | **2.15** | **4.2(m)** | **6.27** | **5.3(m)** | **6.92** | **5.8(m)** | **14.02** | **7.8(m)** |
| GA-2opt | 0.17 | 3.3(m) | 0.43 | 7.6(m) | 0.87 | 29.4(m) | 1.62 | 56.3(m) | 3.35 | 167.6(m) | 7.14 | 309.8(m) |
| GA-2opt+PT | 0.05 | 3.6(m) | 0.08 | 8.0(m) | 0.24 | 29.9(m) | 0.54 | 56.7(m) | 1.46 | 168.1(m) | 6.07 | 310.3(m) |
| GA-2opt+GLEET | 0.00 | 3.5(m) | 0.02 | 7.9(m) | 0.09 | 30.9(m) | 0.33 | 57.8(m) | 0.91 | 171.2(m) | 5.47 | 311.5(m) |
| GA-2opt+BEA | 0.01 | 4.5(m) | 0.07 | 8.9(m) | 0.25 | 30.8(m) | 0.41 | 57.7(m) | 1.03 | 169.0(m) | 5.86 | 311.2(m) |
| GA-2opt+EoH | 0.00 | 3.4(m) | 0.04 | 7.8(m) | 0.27 | 29.7(m) | 0.63 | 56.5(m) | 2.91 | 167.9(m) | 5.83 | 310.0(m) |
| GA-2opt+ReEvo | 0.00 | 3.9(m) | 0.02 | 8.5(m) | 0.16 | 30.2(m) | 0.48 | 57.2(m) | 2.68 | 168.5(m) | 5.95 | 310.7(m) |
| **GA-2opt+AutoEP** | **0.00** | **5.8(m)** | **0.01** | **10.1(m)** | **0.01** | **31.9(m)** | **0.09** | **58.9(m)** | **0.08** | **170.2(m)** | **3.58** | **312.8(m)** |
| GA-2opt+EoH+AutoEP | 0.00 | 5.9(m) | 0.01 | 10.2(m) | 0.01 | 32.5(m) | 0.11 | 59.1(m) | 0.08 | 169.6(m) | 3.61 | 312.6(m) |
| GA-2opt+ReEvo+AutoEP | 0.00 | 6.2(m) | 0.01 | 10.6(m) | 0.01 | 32.1(m) | 0.10 | 59.8(m) | 0.07 | 170.1(m) | 3.59 | 312.3(m) |

For the TSP problem, we selected the TSPLIBReinelt (1991) dataset and used GAHolland (1992) and GA-2opt Sabba & Chikhi (2013) as baseline algorithms. GA is a widely used metaheuristic algorithm, while GA-2opt, which combines global and local search, is a robust heuristic for TSP. Detailed experimental results are presented in Table 1.The first row displays the performance of neural combinatorial optimization methods on various TSP datasets. The second row compares hyperparameter tuning methods and LLMs-enhanced metaheuristic operators using the GA algorithm. Among hyperparameter tuning methods, GLEET performed best, while ReEvo showed promising results in enhancing metaheuristic operators. AutoEP, after optimizing GA's hyperparameters, achieved the best results across all problem sizes.The third row evaluates GA-2opt, which combines population-based and local search strategies, resulting in strong performance. The test results indicate that the algorithm with dynamically controlled hyperparameters by AutoEP achieved SOTA results across all test datasets, surpassing current neural combinatorial optimization SOTA methods like LEHD and DACT. These comparative results demonstrate that AutoEP, as a plug-and-play framework for tuning metaheuristic algorithm hyperparameters, significantly enhances the performance of the original algorithms. To validate AutoEP's ability to dynamically adjust hyperparameters when integrated as a plugin with any metaheuristic algorithm, we applied it to ReEvo and EoH-enhanced GA-2opt algorithms, as shown in the fourth row. Our results demonstrate that AutoEP further improves the performance of these enhanced algorithms, with final results closely matching those of GA-2opt+AutoEP. This confirms two key points:

(1) Online adaptation is crucial: Even a well-designed initial heuristic benefits from dynamic, state-aware control during the run.

(2) AutoEP is a general-purpose enhancer: It acts as a plug-and-play module that can improve any given metaheuristic, including those already enhanced by other methods.

Computational Overhead is Minimal. Our CoR architecture, which leverages efficient 30B-parameter models, is highly practical. The average inference latency per decision is negligible (30 ms). Over an entire optimization run with hundreds of adjustments, the total added time is minimal (e.g., 2-5 minutes on longer runs), a small price for a significant improvement in solution quality.

### 4.2.2 VALIDATION ON CVRP, FSSP, AND UAV TRAJECTORY OPTIMIZATION

A detailed analysis of the experimental results on the CVRP, FSSP, and UAV trajectory optimization datasets can be found in Appendix B.

## 4.3 ABLATION STUDIES: DECONSTRUCTING AUTOEP'S PERFORMANCE

To isolate the contributions of AutoEP's key components, we conducted two ablation studies on the TSP benchmark.

The Criticality of State-Sensing (ELA) and Reasoning (CoR). Table 2 demonstrates that both the ELA module and the CoR engine are essential for effective performance.

**Without ELA**, the LLM lacks situational awareness. Though it can still see past actions and outcomes from the experience pool, its performance degrades significantly, as it is reasoning without a real-time understanding of the search dynamics.

**Without CoR (using a single LLM)**, performance drops to the level of the baseline. This shows that simply feeding raw state features to a standard LLM is insufficient; our structured, decomposed reasoning pipeline is crucial for translating state information into an effective strategy.

**Without both**, the framework operates blindly, leading to chaotic adjustments that perform worse than the untuned baseline.

Efficiency and Accessibility: CoR vs. Monolithic SOTA LLMs. We then investigated whether our multi-LLM CoR could be replaced by a single, powerful, proprietary model (e.g., GPT-o1, Gemini 2.5 Pro). As shown in Table 3, our CoR, built with efficient 30B-class open-source models, achieves performance on par with these massive SOTA models. However, it does so with an order of magnitude less computational time (e.g., 5.8 min vs. 50 min on eil51). This is a critical finding: our structured reasoning framework provides a path to achieving SOTA performance without relying on expensive, slow, and proprietary APIs. It makes advanced, LLM-driven algorithm control practical, accessible, and locally deployable.

Table 2: Component ablation study of AutoEP on TSP.

| Method | eil51 Opt.gap(%)↓ | Rd100 Opt.gap(%)↓ | Kroa150 Opt.gap(%)↓ | rd300 Opt.gap(%)↓ | rat575 Opt.gap(%)↓ | dsj1000 Opt.gap(%)↓ |
|---|---|---|---|---|---|---|
| GA-2opt | 0.17 | 0.43 | 0.87 | 1.62 | 3.35 | 7.14 |
| GA-2opt+AutoEP (Without ELA) | 0.06 | 0.33 | 0.57 | 1.30 | 3.11 | 6.46 |
| GA-2opt+AutoEP (Without CoR) | 0.16 | 0.43 | 0.81 | 1.60 | 3.37 | 7.11 |
| GA-2opt+AutoEP (Without ELA+CoR) | 0.21 | 0.56 | 1.37 | 1.84 | 3.91 | 7.93 |
| **GA-2opt+AutoEP** | **0.00** | **0.01** | **0.01** | **0.09** | **0.08** | **3.58** |

Table 3: Comparison of CoR components with other reasoning LLMs.

| Method | eil51 Opt.gap(%)↓ | Time | Rd100 Opt.gap(%)↓ | Time | Kroa150 Opt.gap(%)↓ | Time | rd300 Opt.gap(%)↓ | Time | rat575 Opt.gap(%)↓ | Time | dsj1000 Opt.gap(%)↓ | Time |
|---|---|---|---|---|---|---|---|---|---|---|---|---|
| AutoEP without CoR(GPT-o1) | 0.00 | 44.7(m) | 0.01 | 49.4(m) | 0.01 | 71.0(m) | 0.09 | 97.9(m) | 0.09 | 209.2(m) | 1.59 | 351.1(m) |
| AutoEP without CoR (Claude 3.7) | 0.00 | 43.6(m) | 0.03 | 47.5(m) | 0.02 | 69.1(m) | 0.11 | 95.4(m) | 0.10 | 203.8(m) | 1.58 | 343.6(m) |
| AutoEP without CoR (Gemini 2.5 Pro) | 0.00 | 51.6(m) | 0.01 | 56.9(m) | 0.01 | 78.2(m) | 0.08 | 105.1(m) | 0.08 | 214.5(m) | 1.57 | 361.1(m) |
| AutoEP without CoR (DeepSeek-R1) | 0.00 | 53.9(m) | 0.01 | 58.2(m) | 0.01 | 81.3(m) | 0.09 | 107.4(m) | 0.11 | 218.6(m) | 1.59 | 363.4(m) |
| **AutoEP with CoR (Qwen3-30B)** | **0.00** | **5.8(m)** | **0.01** | **10.1(m)** | **0.01** | **31.9(m)** | **0.09** | **58.9(m)** | **0.08** | **170.2(m)** | **1.58** | **312.8(m)** |

## 4.4 ROBUSTNESS TO FOUNDATIONAL MODEL CAPABILITIES

A key concern with LLM-based systems is their dependence on the underlying model's power. We tested this by running AutoEP, EoH, and ReEvo with various LLMs. As shown in Figure 4, the performance of EoH and ReEvo, which rely on the LLM's raw generative ability, degrades significantly when using smaller models. In contrast, AutoEP maintains its high performance even with less powerful models. This demonstrates that AutoEP's strength comes from its structured framework (grounding via ELA, reasoning via CoR), not just the raw intelligence of the LLM. This architectural robustness makes AutoEP more reliable and practical for real-world deployment.

## 4.5 SENSITIVITY TO ADJUSTMENT FREQUENCY

We analyzed the trade-off between decision frequency and performance on the UAV problem (Figure 5). While adjusting at every iteration yields the fastest convergence, less frequent adjustments (e.g., every 3-5 iterations) still provide substantial benefits while reducing computational overhead. This provides a practical "knob" for users: on problems with very long runtimes, one can reduce the adjustment frequency to save time without sacrificing the majority of the performance gain.

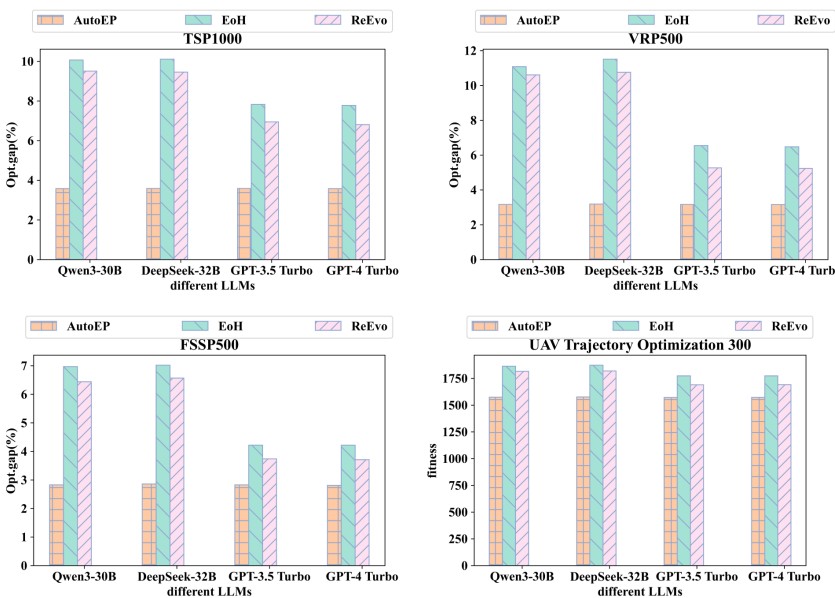

Figure 4: Comparison of Experimental Results Across Different LLMs. The baseline algorithm for adjustment is GA-2opt.

A critical concern in using LLMs for online control is the management of context length. As noted in Sec3.2, the Experience Pool stores historical states and actions. To prevent prompt bloat and minimize context noise, AutoEP employs a **sliding window mechanism**, retaining only the most recent $L$ iterations. We conducted an ablation study on the TSP-100 dataset using GA-2opt to determine the optimal window size. We compared short ($L = 5$), medium ($L = 20$), long ($L = 50$), and infinite (Full History) window sizes.

As shown in Table 4, utilizing the full history significantly degrades both efficiency and solution quality. The performance drop in the "Full History" setting confirms that irrelevant historical data (e.g., early-stage exploration metrics) acts as context noise during later optimization stages, causing the LLM to generate suboptimal strategies. Conversely, a very small window ($L = 5$) is computationally efficient but lacks sufficient temporal context to identify complex trends like stagnation. The default setting of $L = 20$ strikes the optimal balance, providing sufficient history for trend analysis while maintaining low inference latency and filtering out obsolete data.

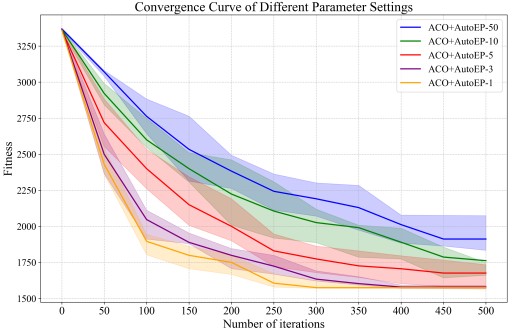

Figure 5: Comparison of hyperparameter tuning frequencies(UAV-300).

Table 4: Impact of experience pool size ($L$) on performance and inference latency (TSP-100).

| Pool Size ($L$) | Opt. Gap (%) | Avg. Inference Latency (s) | Observation |
|---|---|---|---|
| $L = 5$ | 0.04 | **0.18** | Limited context; reactive behavior |
| $L = 20$ (Default) | **0.01** | 0.31 | Balanced context and efficiency |
| $L = 50$ | 0.03 | 0.85 | Increased latency; minor distractions |
| Full History | 0.17 | $> 2.50$ | High latency; hallucination from noise |

## 4.6 INTERPRETABILITY ANALYSIS: VISUALIZING LLM-DRIVEN CONTROL DYNAMICS

To demonstrate the interpretability of AutoEP, we visualize the evolutionary trajectory of GA hyperparameters (Mutation and Crossover probabilities) on the TSP-400 instance (Figure 6). The resulting

curves reveal a sophisticated, state-aware control strategy that distinctively contrasts with manual, monotonic schedules.

**Response to ELA Features (Attention to State)**: The LLM's decisions are explainable through the observed ELA metrics. Periods of rapid parameter fluctuation correspond to specific landscape characteristics. For example, when the Dispersion Ratio ($D_{ratio}$) indicates a single-funnel structure4, AutoEP prioritizes exploitation. Conversely, when Skewness signals a risk of premature convergence5, the system triggers a 'rescue' behavior by sharply increasing the mutation rate.

**Operationalizing the Exploration-Exploitation Trade-off**: The visualization confirms that AutoEP understands the mechanics of GA. The crossover and mutation curves often mirror each other (inverse correlation). This validates that the CoR engine successfully translates abstract directives (e.g., 'intensify exploration') into logically consistent numerical actions (High Mutation / Low Crossover).

**Emergence of Search Policies via In-Context Learning**: Beyond direction, the magnitude of adjustments shows structured patterns. The LLM utilizes the Experience Pool to determine not just what to change, but how much. This results in distinct phases: aggressive large-step adjustments to escape stagnation, followed by fine-grained small-step tuning for local convergence, proving the effective utilization of historical search context.

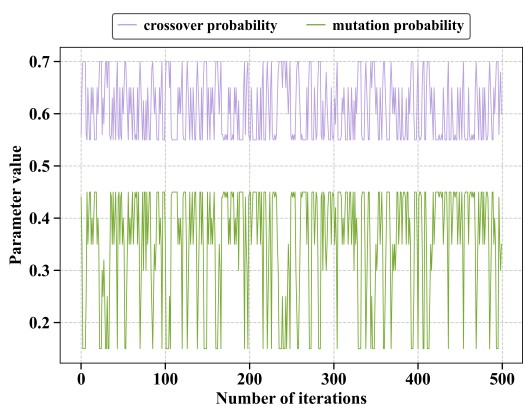

Figure 6: Visualization of hyperparameter evolution for GA on TSP-400.

## 5 DISCUSSION AND CONCLUSION

In this work, we introduced AutoEP, a framework that pioneers a new paradigm for automated algorithm configuration. By synergizing real-time search analytics (ELA) with the reasoning capabilities of LLMs, we have demonstrated a system that can dynamically control complex metaheuristic algorithms in a zero-shot, training-free manner. Our extensive experiments show that this approach not only outperforms state-of-the-art hyperparameter tuners but also elevates classical heuristics to a performance level competitive with specialized neural optimization methods.

**Broader Implications: A Shift from Learning to Reasoning.** Our work represents a fundamental departure from the dominant "learning from scratch" paradigm, exemplified by reinforcement learning. Instead of investing massive computational resources to train a control policy for every new problem or algorithm variant, AutoEP leverages the rich prior knowledge embedded within pre-trained LLMs. This transforms the problem of algorithm control from one of sample-intensive learning to one of efficient, in-context reasoning. At the core of AutoEP is a "sense-reason-act" loop, where ELA provides the senses, the CoR provides the reasoning, and hyperparameter adjustments are the actions. This loop offers a generalizable blueprint for creating more adaptive and intelligent computational systems.

**Practical Advantages for Real-World Optimization.** Beyond its novelty, AutoEP is designed for practicality. Its plug-and-play framework makes it a general-purpose tool for enhancing any metaheuristic algorithm. Crucially, as demonstrated by our ablation studies, AutoEP's structured reasoning framework reduces the dependency on a single, monolithic LLM's raw intelligence. This architectural strength allows it to achieve SOTA performance using smaller, open-source models (e.g., 30B-32B class). This is a critical advantage for real-world applications like factory scheduling or logistics, where local deployment is necessary to ensure data privacy, low latency, and operational reliability, and where deploying massive proprietary models is often infeasible.

## ETHICAL STATEMENT AND REPRODUCIBILITY STATEMENT

Our paper has no conflicts of interest and complies with ethical standards. Our paper code is reproducible, and we have provided an anonymous link to the reproducible code.

ACKNOWLEDGMENTS

This work was supported in part by the National Natural Science Foundation of China under Grant Nos. 62273352, 72501290, and 72401287. The authors would like to thank the anonymous reviewers for their constructive feedback, which helped improve the quality of this paper.

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

## A  LLM USAGE

We only used LLMs to polish the paper writing.

## B  DETAILED EXPERIMENTAL RESULTS

### B.1  VALIDATION ON CVRP

For the CVRP problem, we used the VRPLIB Semet & Taillard (1993) dataset and included both GA-2opt and PSO-2opt Sabba & Chikhi (2013) algorithms to evaluate AutoEP's performance across different metaheuristic algorithms. The test results are presented in Table 5.In the first column, DACT and LEHD continue to show strong performance. The second column compares various methods for improving PSO-2opt, where AutoEP achieves superior results. The third column evaluates enhancements to GA-2opt, with AutoEP demonstrating the best performance. Additionally, GA-2opt combined with AutoEP achieves the smallest gap from the optimal solutions across all datasets.

Table 5: Comparison with various baselines on CVRP. Opt.gap represents the percentage gap between the average run result and the optimal solution; a smaller value is better. Time is the average runtime (unit: minute).

| Method | N=20 | | N=50 | | N=100 | | N=200 | | N=500 | |
|---|---|---|---|---|---|---|---|---|---|---|
| | Opt.gap(%)↓ | Time | Opt.gap(%)↓ | Time | Opt.gap(%)↓ | Time | Opt.gap(%)↓ | Time | Opt.gap(%)↓ | Time |
| DACT | 0.01 | 0.5(m) | 0.09 | 3.8(m) | 0.57 | 7.2(m) | 3.45 | 16.5(m) | 7.52 | 24.0(m) |
| LEHD | 0.01 | 0.2(m) | 0.13 | 0.2(m) | 0.64 | 0.3(m) | 3.61 | 0.4(m) | 7.64 | 0.7(m) |
| PSO-2opt | 1.38 | 3.0(m) | 1.65 | 7.5(m) | 2.33 | 28.0(m) | 6.53 | 55.0(m) | 9.66 | 165.0(m) |
| PSO-2opt+PT | 0.19 | 3.3(m) | 0.29 | 7.9(m) | 1.13 | 28.4(m) | 3.77 | 55.3(m) | 5.81 | 165.4(m) |
| PSO-2opt+GLEET | 0.08 | 3.8(m) | 0.14 | 9.2(m) | 0.97 | 30.2(m) | 2.93 | 57.1(m) | 4.82 | 167.2(m) |
| PSO-2opt+BEA | 0.11 | 4.1(m) | 0.26 | 8.7(m) | 1.04 | 29.3(m) | 3.59 | 56.2(m) | 5.31 | 166.3(m) |
| PSO-2opt+EoH | 0.12 | 3.2(m) | 0.33 | 7.7(m) | 1.30 | 28.3(m) | 4.71 | 55.2(m) | 7.47 | 165.2(m) |
| PSO-2opt+ReEvo | 0.09 | 3.7(m) | 0.27 | 8.3(m) | 1.27 | 28.8(m) | 3.92 | 55.8(m) | 6.41 | 165.8(m) |
| **PSO-2opt+AutoEP** | **0.06** | **5.8(m)** | **0.09** | **10.7(m)** | **0.83** | **31.2(m)** | **2.48** | **58.6(m)** | **4.25** | **168.5(m)** |
| GA-2opt | 0.91 | 3.5(m) | 1.03 | 8.0(m) | 1.88 | 31.0(m) | 5.89 | 59.0(m) | 8.1 | 178.0(m) |
| GA-2opt+PT | 0.26 | 3.8(m) | 0.20 | 8.4(m) | 0.59 | 31.5(m) | 1.93 | 59.4(m) | 5.93 | 178.5(m) |
| GA-2opt+GLEET | 0.01 | 3.7(m) | 0.07 | 8.2(m) | 0.19 | 31.8(m) | 1.44 | 59.7(m) | 4.07 | 178.8(m) |
| GA-2opt+BEA | 0.07 | 4.6(m) | 0.11 | 9.1(m) | 0.24 | 32.1(m) | 1.63 | 60.1(m) | 4.71 | 179.1(m) |
| GA-2opt+EoH | 0.08 | 3.6(m) | 0.15 | 8.2(m) | 0.63 | 31.2(m) | 2.17 | 59.2(m) | 6.55 | 178.2(m) |
| GA-2opt+ReEvo | 0.03 | 4.0(m) | 0.08 | 8.6(m) | 0.44 | 31.6(m) | 1.69 | 59.6(m) | 5.27 | 178.6(m) |
| **GA-2opt+AutoEP** | **0.01** | **6.1(m)** | **0.05** | **10.9(m)** | **0.13** | **33.9(m)** | **1.08** | **62.1(m)** | **3.17** | **181.1(m)** |
| GA-2opt+EoH+AutoEP | 0.01 | 6.3(m) | 0.06 | 11.2(m) | 0.13 | 34.2(m) | 1.09 | 62.3(m) | 3.17 | 181.4(m) |
| GA-2opt+ReEvo+AutoEP | 0.01 | 6.2(m) | 0.05 | 11.1(m) | 0.14 | 34.1(m) | 1.07 | 62.4(m) | 3.15 | 181.3(m) |

### B.2  VALIDATION ON FSSP

For the FSSP problem, we used the Taillard Taillard (1993) dataset and GA-2opt as the baseline algorithm. We also included advanced methods for FSSP: NEH Nawaz et al. (1983), NEHFF Fernandez-Viagas & Framinan (2014), and PFSPNet_NEH Pan et al. (2021). The test results are presented in Table 6. In the first column, PFSPNet_NEH shows superior performance among the comparison algorithms. The second column demonstrates that AutoEP significantly enhances GA-2opt across all datasets, consistently achieving the best results.

### B.3  VALIDATION ON UAV TRAJECTORY OPTIMIZATION

In remote or disaster-stricken areas where ground-based network connectivity is unavailable, using UAVs for data collection and transmission has become a significant research focus Yan et al. (2023). Testing AutoEP's performance in more complex optimization scenarios is therefore highly relevant. UAV trajectory optimization for data collection involves factors such as flight speed, energy consumption due to environmental resistance, data collection rate, and storage capacity. The optimization goal is to minimize data collection time. The ACO Dorigo et al. (2007) algorithm, widely used in trajectory optimization, was chosen as the baseline for comparison.A detailed mathematical model is presented in Liu et al. (2022). Experimental results, presented in Table 7, show the minimized data collection times for varying sensor node numbers. Compared to other methods that improve ACO, AutoEP demonstrated the greatest enhancement, improving ACO's performance by 17.16% with 300 sensor nodes.

Table 6: Comparison with various baselines on FSSP. Opt.gap represents the percentage gap between the average run result and the optimal solution; a smaller value is better. Time is the average runtime (unit: minute).

| Method | n20m10 | | N50m10 | | N100m20 | | N200m20 | | N500m20 | |
|---|---|---|---|---|---|---|---|---|---|---|
| | Opt.gap(%)↓ | Time | Opt.gap(%)↓ | Time | Opt.gap(%)↓ | Time | Opt.gap(%)↓ | Time | Opt.gap(%)↓ | Time |
| NEH | 4.05 | 0.4(m) | 3.47 | 0.7(m) | 3.58 | 1.8(m) | 5.27 | 3.8(m) | 4.59 | 4.7(m) |
| NEHFF | 4.15 | 0.5(m) | 3.62 | 0.8(m) | 3.73 | 2.0(m) | 5.82 | 4.0(m) | 4.83 | 4.9(m) |
| PFSPNet | 4.04 | 0.6(m) | 3.48 | 0.9(m) | 3.56 | 2.2(m) | 6.05 | 4.2(m) | 5.36 | 5.0(m) |
| GA-2opt | 4.37 | 3.6(m) | 5.15 | 8.2(m) | 6.42 | 31.5(m) | 5.62 | 60.0(m) | 7.83 | 178.0(m) |
| GA-2opt+PT | 3.16 | 3.9(m) | 3.70 | 8.6(m) | 4.19 | 32.0(m) | 3.93 | 60.4(m) | 4.09 | 178.5(m) |
| GA-2opt+GLEET | 2.64 | 4.1(m) | 2.95 | 10.5(m) | 3.67 | 33.8(m) | 3.28 | 62.3(m) | 3.52 | 180.2(m) |
| GA-2opt+BEA | 2.91 | 4.5(m) | 3.36 | 9.3(m) | 3.95 | 32.6(m) | 3.53 | 61.2(m) | 3.81 | 179.3(m) |
| GA-2opt+EoH | 3.31 | 3.7(m) | 3.87 | 8.4(m) | 4.43 | 31.8(m) | 3.64 | 60.2(m) | 4.22 | 178.3(m) |
| GA-2opt+ReVo | 2.85 | 4.0(m) | 3.16 | 8.9(m) | 3.88 | 32.2(m) | 3.31 | 60.7(m) | 3.74 | 178.8(m) |
| **GA-2opt+AutoEP** | **2.09** | **6.3(m)** | **2.80** | **10.8(m)** | **3.16** | **34.6(m)** | **2.93** | **63.2(m)** | **2.83** | **181.5(m)** |
| GA-2opt+EoH+AutoEP | 2.08 | 6.5(m) | 2.81 | 11.0(m) | 3.16 | 34.9(m) | 2.96 | 63.5(m) | 2.85 | 181.8(m) |
| GA-2opt+ReEvo+AutoEP | 2.08 | 6.4(m) | 2.80 | 10.9(m) | 3.14 | 34.7(m) | 2.93 | 63.3(m) | 2.81 | 181.6(m) |

Table 7: Comparison of UAV Trajectory Optimization Experiments. Traj.Length is the length of the drone's flight trajectory, where a lower value indicates a better performance. Time is the average runtime (unit: minute).

| Method | n20 | | N50 | | N100 | | N200 | | N300 | |
|---|---|---|---|---|---|---|---|---|---|---|
| | Traj.Length↓ | Time | Traj.Length↓ | Time | Traj.Length↓ | Time | Traj.Length↓ | Time | Traj.Length↓ | Time |
| ACO | 147.33 | 2.2(m) | 312.23 | 4.0(m) | 607.29 | 20.0(m) | 1387.05 | 38.5(m) | 1912.74 | 90.0(m) |
| ACO+PT | 133.04 | 2.5(m) | 297.73 | 4.4(m) | 576.18 | 20.5(m) | 1182.78 | 39.0(m) | 1713.64 | 90.5(m) |
| ACO+GLEET | 129.86 | 2.7(m) | 295.97 | 6.0(m) | 564.91 | 22.8(m) | 1125.31 | 41.3(m) | 1683.40 | 92.8(m) |
| ACO+BEA | 131.70 | 2.6(m) | 297.63 | 5.2(m) | 572.84 | 21.2(m) | 1146.79 | 39.7(m) | 1706.41 | 91.2(m) |
| ACO+EoH | 136.41 | 2.4(m) | 302.54 | 4.3(m) | 587.46 | 20.3(m) | 1208.19 | 38.8(m) | 1774.25 | 90.3(m) |
| ACO+ReEvo | 131.56 | 2.5(m) | 297.46 | 4.7(m) | 571.26 | 20.9(m) | 1184.46 | 39.2(m) | 1690.83 | 90.9(m) |
| **ACO+AutoEP** | **122.08** | **4.2(m)** | **291.58** | **6.5(m)** | **550.31** | **23.0(m)** | **1079.83** | **41.7(m)** | **1574.90** | **93.5(m)** |
| ACO+EoH+AutoEP | 122.09 | 4.5(m) | 291.61 | 6.9(m) | 550.33 | 23.4(m) | 1079.87 | 42.0(m) | 1574.87 | 93.8(m) |
| ACO+ReEvo+AutoEP | 122.06 | 4.3(m) | 291.58 | 6.7(m) | 550.30 | 23.2(m) | 1079.82 | 41.9(m) | 1574.92 | 93.6(m) |

# C  PROMPT

## C.1  PROMPT FOR THE GA

Below is the complete set of prompts used for the GA algorithm in solving the TSP problem:

> *In genetic algorithms, hyperparameters such as crossover probability and mutation probability play a critical role. Could you analyze the impact of these hyperparameters on the algorithm's exploration-exploitation balance?*

Figure 7: GA:Prompt for Strategist LLM.

## C.2  PROMPT FOR THE PSO

Below is the complete set of prompts for the PSO algorithm in solving the CVRP problem:

> *In the PSO algorithm, key hyperparameters include inertia weight, social learning factor, and individual learning factor. Could you analyze how these hyperparameters affect the exploration-exploitation balance of the algorithm?*

Figure 10: PSO: Prompt for Strategist LLM.

*The experience pool contains data from previous iterations:*
*Iterations: 10,Crossover Probability: 0.7, Mutation Probability: 0.1, Best Fitness Value: 1168.*
*Current algorithm state features: Kurtosis: 0.3, Skewness: 0.27, Diversity: 1, $R^2$:0.13, $D_{ratio}$:0.78.*
*Feature descriptions:*
*1. If the skewness value is close to 0, the solution distribution is symmetric, indicating a balance between exploration and exploitation. If the skewness is significantly greater than 0, most solutions are poor, suggesting the need for more local search and exploitation. If skewness is significantly less than 0, there is a risk of converging to a local optimum, necessitating an increase in exploration.*
*2. If the kurtosis is near 0, the fitness values are balanced between the mean and the tails. When the kurtosis is significantly greater than 0, the solutions are concentrated, and diversity is low, requiring an increase in exploration. Conversely, when the kurtosis is significantly less than 0, the solution set is dispersed, suggesting a need to search for local optima, thus increasing exploitation.*
*3. If the diversity value is greater than 1, the population is still evolving, and increasing local search could improve exploitation. If the diversity is less than or equal to 1, the population is stuck, and increasing exploration is necessary.*
*4. A high $R^2 \approx 1$ indicates a well-structured landscape (i.e., a funnel), signaling a need to increase exploitation. Conversely, a low $R^2 \approx 0$ implies a rugged or multi-modal landscape, requiring an increase in exploration to avoid premature convergence.*
*5. A value of $D_{ratio} \ll 1$ (e.g., < 0.2) is a strong indicator of a single funnel structure, as the best solutions are tightly clustered. This signals the need to increase exploitation to refine the search within this promising basin. Conversely, a value of $D_{ratio} \approx 1$ suggests a multi-modal landscape, where elite solutions are found in disparate regions. This necessitates an increase in exploration to avoid premature convergence to a local optimum.*

*Based on feature descriptions and historical decision information, determine whether the algorithm requires further exploration or development. No additional output is required.*

Figure 8: GA: Prompt for Analyst LLM.

*The current hyperparameters of the genetic algorithm are:*
*Crossover Probability: 0.7,Mutation Probability: 0.1. The range of the crossover probability is [0.55, 0.7], and the range of the mutation probability is [0.1, 0.45]. In genetic algorithms, crossover probability promotes development by controlling gene exchange, while mutation probability enhances exploration by introducing random disruptions to search new solution spaces. Based on the algorithm state analysis, there is a need to increase exploration. Please analyze how to adjust hyperparameters by referencing historical decision information. No additional output is required; only the specific values of the hyperparameters need to be provided.*

Figure 9: GA: Prompt for Actuator LLM.

---

*The experience pool contains data from previous iterations:*
*Iterations: 20, Inertia Weight: 0.3, Social Learning Factor: 2.2, Individual Learning Factor: 1.5,Best Fitness Value: 790.Current algorithm state features: Kurtosis: 0.22, Skewness: -0.35, Diversity: 1.3, $R^2$:0.85, $D_{ratio}$:0.14.*
*Feature descriptions:*
*1. If the skewness value is close to 0, the solution distribution is symmetric, indicating a balance between exploration and exploitation. If the skewness is significantly greater than 0, most solutions are poor, suggesting the need for more local search and exploitation. If skewness is significantly less than 0, there is a risk of converging to a local optimum, necessitating an increase in exploration.*
*2. If the kurtosis is near 0, the fitness values are balanced between the mean and the tails. When the kurtosis is significantly greater than 0, the solutions are concentrated, and diversity is low, requiring an increase in exploration. Conversely, when the kurtosis is significantly less than 0, the solution set is dispersed, suggesting a need to search for local optima, thus increasing exploitation.*
*3. If the diversity value is greater than 1, the population is still evolving, and increasing local search could improve exploitation. If the diversity is less than or equal to 1, the population is stuck, and increasing exploration is necessary.*
*4. A high $R^2 \approx 1$ indicates a well-structured landscape (i.e., a funnel), signaling a need to increase exploitation. Conversely, a low $R^2 \approx 0$ implies a rugged or multi-modal landscape, requiring an increase in exploration to avoid premature convergence.*
*5. A value of $D_{ratio} \ll 1$ (e.g., < 0.2) is a strong indicator of a single funnel structure, as the best solutions are tightly clustered. This signals the need to increase exploitation to refine the search within this promising basin. Conversely, a value of $D_{ratio} \approx 1$ suggests a multi-modal landscape, where elite solutions are found in disparate regions. This necessitates an increase in exploration to avoid premature convergence to a local optimum.*

*Based on feature descriptions and historical decision information, determine whether the algorithm requires further exploration or development. No additional output is required.*

---

Figure 11: PSO: Prompt for Analyst LLM.

---

*The current hyperparameters for the PSO algorithm are:*
*Inertia Weight: 0.3, Social Learning Factor: 2.2, Individual Learning Factor: 1.5. The range of the inertia weight is [0.2, 1]. In the PSO algorithm, increasing inertia weight enhances exploration but weakens exploitation, while decreasing inertia weight strengthens exploitation but reduces exploration. Increasing the individual learning factor boosts exploration, whereas increasing the social learning factor enhances exploitation. Given the algorithm's need for increased exploitation. Please analyze how to adjust hyperparameters by referencing historical decision information. No additional output is required; only the specific values of the hyperparameters need to be provided.*

---

Figure 12: PSO: Prompt for Actuator LLM.

## C.3 PROMPT FOR THE ACO

Below is the complete set of prompts for the ACO algorithm in solving the UAV Trajectory Optimization problem:

---

*In the Ant Colony Optimization algorithm, key hyperparameters include the pheromone factor, heuristic factor, and pheromone evaporation rate. Could you analyze how these hyperparameters affect the exploration-exploitation balance of the algorithm?*

---

Figure 13: ACO: Prompt for Strategist LLM.

---

*The experience pool contains data from previous iterations:*
*Iterations: 90, Pheromone Factor: 1.8, Heuristic Factor: 1.3, Pheromone Evaporation Rate: 0.2,Best Fitness Value: 437.Current algorithm state features: Kurtosis: 0.02, Skewness: 0.02, Diversity: 1.1, $R^2$:0.47, $D_{ratio}$:0.53.*
*Feature descriptions:*
*1. If the skewness value is close to 0, the solution distribution is symmetric, indicating a balance between exploration and exploitation. If the skewness is significantly greater than 0, most solutions are poor, suggesting the need for more local search and exploitation. If skewness is significantly less than 0, there is a risk of converging to a local optimum, necessitating an increase in exploration.*
*2. If the kurtosis is near 0, the fitness values are balanced between the mean and the tails. When the kurtosis is significantly greater than 0, the solutions are concentrated, and diversity is low, requiring an increase in exploration. Conversely, when the kurtosis is significantly less than 0, the solution set is dispersed, suggesting a need to search for local optima, thus increasing exploitation.*
*3. If the diversity value is greater than 1, the population is still evolving, and increasing local search could improve exploitation. If the diversity is less than or equal to 1, the population is stuck, and increasing exploration is necessary.*
*4. A high $R^2 \approx 1$ indicates a well-structured landscape (i.e., a funnel), signaling a need to increase exploitation. Conversely, a low $R^2 \approx 0$ implies a rugged or multi-modal landscape, requiring an increase in exploration to avoid premature convergence.*
*5. A value of $D_{ratio} \ll 1$ (e.g., < 0.2) is a strong indicator of a single funnel structure, as the best solutions are tightly clustered. This signals the need to increase exploitation to refine the search within this promising basin. Conversely, a value of $D_{ratio} \approx 1$ suggests a multi-modal landscape, where elite solutions are found in disparate regions. This necessitates an increase in exploration to avoid premature convergence to a local optimum.*

*Based on the description of the features, determine whether the algorithm needs more exploration or exploitation. No additional output is required.*

Figure 14: ACO: Prompt for Analyst LLM.

---

*The current hyperparameters for the ACO algorithm are:*
*Pheromone Factor: 1.8, Heuristic Factor: 1.3, Pheromone Evaporation Rate: 0.2. The pheromone factor and heuristic factor ranges are [1, 4], and the pheromone evaporation rate range is [0.1, 0.9]. In the ACO algorithm, increasing the pheromone factor and heuristic factor enhances exploitation, while decreasing them increases exploration. Increasing the pheromone evaporation rate promotes exploration. Based on the current algorithm state analysis, the goal is to maintain a balance. Please analyze how to adjust hyperparameters by referencing historical decision information. No additional output is required.*

Figure 15: ACO: Prompt for Actuator LLM.

## D  DETAILS OF THE EXPERIMENTAL SETUP

All LLMs used in this study were accessed via the publicly available APIs provided by their respective developers. The experiments were conducted on a system equipped with an Intel Core i9-13900K CPU and an NVIDIA A800*4 GPU. On average, adjusting hyperparameters using AutoEP took 0.3 seconds per run. For performance evaluation, the LLM employed by AutoEP was Qwen3-30B, while the LLMs used in other methods followed the configurations specified in the original papers. The configurations for the comparison algorithms were strictly adhered to as outlined in the original studies. The parameter settings for the improved base algorithm are shown in Table 8.

Table 8: Parameterization of each meta - heuristic algorithm

| Algorithm | Parameter | Value |
|---|---|---|
| GA | Population size | 500 |
| | Maximum number of iterations | 500 |
| | Initial crossover probability | 0.6 |
| | Initial mutation probability | 0.1 |
| PSO | Population size | 500 |
| | Maximum number of iterations | 500 |
| | Initial individual learning factor | 1.5 |
| | Initial social learning factor | 1.5 |
| | Inertia weights | 0.3 |
| ACO | Population size | 500 |
| | Maximum number of iterations | 500 |
| | Initial pheromone factor | 2 |
| | Initial heuristic factor | 2 |
| | Initial pheromone volatility factor | 0.3 |

