# OpenReview forum: "AutoEP: LLMs-Driven Automation of Hyperparameter Evolution for Metaheuristic Algorithms"
_ICLR.cc/2026/Conference — ICLR 2026 Oral_

### Official Review · Reviewer_xA6g · 2025-10-31

**Soundness:** 3
**Presentation:** 3
**Contribution:** 3
**Rating:** 6
**Confidence:** 4

**Summary:**

This paper introduces a new framework called AutoEP, which aims to optimize the performance of metaheuristic algorithms by using LLMs to automatically adjust the algorithm's hyperparameters. The main contribution of the paper is the proposal of a novel framework that does not require training and can dynamically adjust the algorithm's hyperparameters through the zero-shot reasoning ability of LLMs.

**Strengths:**

1) The paper is organized and easy to follow, with a clear presentation of the problem, methodology, and results.
2) The experiments are thorough, using multiple benchmarks and reliable comparisons to validate the proposed method.
3) The method of using LLMs for zero-shot reasoning to adjust hyperparameters is novel and offers a fresh perspective in automated algorithm design.

**Weaknesses:**

1) Dependency on LLM Performance: AutoEP’s effectiveness relies on the capabilities of the underlying LLM, which may vary in accuracy and efficiency, affecting consistency.
2) The current tests focus on standard benchmarks. Broader testing on more complex or dynamic problems would offer better insights into its robustness and scalability. Some Case study will be good.

**Questions:**

Just a kind discussion:
In this framework, do you think the algorithm relies more on the reasoning ability of the LLM, the experience (historical knowledge of the search space), or memory (rote learning)?
My essential question is whether the algorithm can solve a completely new combinatorial optimization task (one that is not TSP, for example) by solely depending on its reasoning ability.

---

> ### Author Response · Authors · 2025-11-23
>
> We express our sincere gratitude to the reviewer for their positive assessment of our work. We particularly appreciate the recognition of the novelty of our zero-shot reasoning framework and the rigor of our experimental validation.  Below, we address the specific concerns regarding LLM dependency and generalizability.
>
> ## W1  Dependency on LLM Performance
>
> We acknowledge the concern regarding model dependency. However, a core design principle of AutoEP is to minimize reliance on intrinsic model capabilities via our CoR and ELA modules.
>
> 1. Task Simplification: Unlike code-generation baselines (e.g., EoH, ReEvo) that demand the high precision of proprietary models like GPT-4, AutoEP frames optimization as structured decision-making. This significantly lowers the reasoning threshold required of the backend LLM.
>
> 2. Model Agnosticism (Sec. 4.4): Our experiments confirm this robustness. While generative baselines suffer severe degradation with smaller models, AutoEP maintains SOTA performance even with lightweight open-source backends (e.g., Qwen3-30B, DeepSeek-32B). This demonstrates that our performance drives from the structured "Perception-Reasoning-Action" architecture rather than raw model scale.
>
> ## W2 Need for Broader Testing and Case Studies
>
> We appreciate this constructive suggestion and fully concur that validating performance on complex, dynamic problems is essential. To address this, we have extended our evaluation beyond standard benchmarks (e.g., TSP, CVRP, FSSP) to include a real-world, complex case study: UAV trajectory optimization for IoT data collection.
>
> 1. Problem Complexity: As detailed in Section 4.2.2 and Appendix B.3, this scenario is significantly more intricate than the standard TSP. It incorporates dynamic constraints, including flight velocity, environmental drag (energy expenditure), data acquisition rates, and storage limitations.
>
> 2. Performance Gains: In this realistic setting, AutoEP achieved a 17.16% performance gain over the baseline ACO algorithm (on 300 sensor nodes), significantly outperforming other hyperparameter tuning methods. We believe this constitutes the robust case study requested by the reviewer, demonstrating AutoEP's effectiveness in non-standard, real-world optimization tasks.
>
> ##  Question: Reasoning vs. Experience vs. Memory
>
> We genuinely appreciate this insightful question. It touches upon the fundamental mechanism of AutoEP and its potential for Artificial General Intelligence (AGI) in optimization.
>
> Short Answer: In the AutoEP framework, the algorithm relies primarily on the Reasoning ability of the LLM for strategic direction, supported by short-term Experience (In-context Learning) for precise calibration. It does not rely on Memory (rote learning) of specific problem instances. Therefore, yes, the algorithm is capable of solving completely new combinatorial optimization tasks solely depending on its reasoning ability, provided the task can be modeled by a metaheuristic.
>
> Detailed Analysis:
>
> 1. Reasoning as the Core Driver (Ensuring Generalization): AutoEP leverages the reasoning capability of LLMs to bridge the gap between "Algorithm State" and "Control Strategy."
>
> - Abstraction via ELA: The LLM does not interact with the raw problem data (e.g., city coordinates in TSP or sensor locations in UAV tasks). Instead, it reasons based on ELA features (e.g., fitness landscape ruggedness $R^2$, population diversity $D_{ratio}$).
>
> - Universal Heuristic Logic. The inferences drawn by the Analyst LLM—e.g., "high kurtosis signals stagnation, necessitating increased mutation"—represent fundamental metaheuristic principles. Since these principles are invariant across problem domains (whether TSP, Scheduling, or novel black-box functions), AutoEP achieves zero-shot adaptation on unseen tasks.
>
> 2. Experience for Calibration (Not Rote Learning): The role of the Experience Pool is not to provide a database of solved problems (rote learning), but to facilitate Online In-Context Learning.
>
> - The Actuator LLM uses the history of the current run to calibrate the magnitude of parameter adjustments (e.g., "how much should I increase mutation based on the last attempt?"). This is an adaptive mechanism to fit the specific landscape of the new task dynamically, rather than recalling a fixed solution from memory.
>
> 3. Evidence on "New Tasks": Our experiments explicitly validate this capability. We tested AutoEP on a UAV Trajectory Optimization problem (Section 4.2.2), which involves complex dynamic constraints significantly different from standard benchmarks like TSP or CVRP. AutoEP successfully optimized the ACO algorithm for this task without any task-specific retraining, outperforming baselines by 17.16%. This confirms that AutoEP can indeed tackle completely new tasks by relying on its reasoning architecture.

---

> > ### Comment · Reviewer_xA6g · 2025-11-27
> >
> > Thank you for your detailed response; after careful consideration, I have decided to maintain my original score.

---

### Official Review · Reviewer_yTaP · 2025-10-31

**Soundness:** 3
**Presentation:** 3
**Contribution:** 3
**Rating:** 6
**Confidence:** 4

**Summary:**

This paper proposes AutoEP, an LLM-driven framework for automating hyperparameter evolution in metaheuristic algorithms.The core idea is to replace static control rules or reinforcement-learning-based policies with quantitative reasoning driven by LLMs, thereby enabling adaptive, interpretable hyperparameter control without additional training.AutoEP incorporates an exploratory landscape analysis (ELA) module that extracts numerical descriptors from the ongoing optimization process, and a chain-of-reasoning (CoR) framework consisting of three agents (Strategist, Analyst, Actuator) to translate ELA features into parameter updates.
Experimental results across multiple metaheuristic algorithms (GA, PSO, ACO) and optimization problems (TSP, CVRP, FSSP, UAV trajectory optimization) demonstrate that AutoEP improves performance and runtime efficiency compared to classical adaptive heuristics and recent LLM-enhanced approaches.
Ablation studies further support the importance of ELA grounding and CoR decomposition.

**Strengths:**

1. Introducing an ELA-grounded, LLM-reasoning framework for online parameter control is conceptually new and methodologically sound.
2. The CoR structure (Strategist–Analyst–Actuator) provides interpretability and modularity while reducing reliance on very large proprietary models.
3. The paper evaluates across four distinct problem domains and three algorithmic backbones, demonstrating versatility. Ablation and sensitivity analyses are included and substantiate the key design choices.

**Weaknesses:**

1. Section 3.2 states that all historical states, actions, and outcomes are stored in the Experience Pool.While this guarantees completeness, it also raises concerns about prompt bloat and redundant context as optimization proceeds over hundreds of generations.In large-scale or long-horizon runs, this could degrade inference efficiency and increase context noise for the LLM.
2. The framework claims that the CoR pipeline ensures stable exploration–exploitation balancing, but no mathematical or empirical justification is provided for convergence.
3. Visualizing how LLM decisions evolve (e.g., parameter adjustment curves or attention to specific ELA features) would substantiate the interpretability claim.
4. AutoEP is compared to ReEvo and EoH but not to more recent LLM-based black-box optimizers such as EvoLLM.

**Questions:**

1. Since all historical records are kept, how large does the Experience Pool grow during long optimization runs, and how is memory or context length managed?
2. Is there any mechanism (e.g., damping or adaptive scaling) to prevent over-adjustment of hyperparameters?
3. Can AUTOEP adapt to new algorithm families (e.g., DE or CMA-ES) without additional human intervention?
4. Could you report sensitivity analyses for Experience Pool size and CoR decision frequency?
5. Could you also report per-component latency breakdown (Strategist/Analyst/Actuator) to substantiate the 30 ms decision claim?

---

> ### Author Response · Authors · 2025-11-23
> **Official Comment by Authors(1/2)**
>
> We thank the reviewers for their constructive feedback and for recognizing the novelty of AutoEP in synergizing Exploratory Landscape Analysis (ELA) with Large Language Models for zero-shot algorithm control. We are encouraged by the assessment that our framework is "conceptually new," "methodologically sound," and "versatile." Below, we address the specific concerns regarding context management, convergence, and comparative baselines.
>
> ## W1&Q1 Experience Pool Management & Context Length
>
> **Implementation Clarification**: In our actual implementation (and reported experiments), AutoEP employs a sliding window mechanism, retaining only the most recent $N$ iterations (default $N=20$). We apologize for the omission of this specific truncation strategy in the original manuscript.
>
>
> **New Sensitivity Analysis**: To quantitatively address this concern, we conducted a new sensitivity analysis on the experience pool size ($L \in \{5, 20, 50, \text{Full}\}$), evaluating its impact on both optimization performance and latency. Our key findings are as follows:
>
> 1. Efficiency: As predicted by the reviewer, unbounded history significantly inflates inference time. A window size of $L=20$ maintains low latency ($\approx 0.3$s per decision), whereas using the "Full" history causes substantial delays in later stages.
>
> 2. Performance & Context Noise: We observed that retaining the full history actually degrades performance, confirming the reviewer's concern about "context noise." During the convergence phase, early historical data (e.g., initial random exploration) becomes irrelevant or misleading. This outdated context biases LLM reasoning and increases the risk of hallucination due to excessive input length
>
> 3. Optimal Trade-off: $L=20$ yields the optimal balance. An extremely short window ($L=5$) is too myopic to capture trend dynamics, whereas $L=20$ provides sufficient trajectory history without introducing noise.
>
> These results have been incorporated into the Sensitivity Analysis (Section 4.5) of the revised manuscript.
>
>
> ## W2 Convergence
>
> You raise a significant and challenging theoretical question. We acknowledge that providing a formal mathematical convergence proof for such a highly non-linear, LLM-guided meta-heuristic system remains an open research challenge and falls beyond the scope of this study.
>
> Instead, our contribution is anchored in rigorous empirical validation. We demonstrate that AutoEP achieves stable and efficient convergence through extensive experimentation:
>
> - Cross-Domain Generalizability: Validated across four distinct problem domains: TSP, CVRP, FSSP, and UAV path planning.
>
> - Algorithmic Versatility: Proven effective on three diverse baseline algorithms: GA, PSO, and ACO.
>
> - Superiority over SOTA: AutoEP consistently and significantly outperforms both traditional tuners (e.g., GLEET, BEA) and recent LLM-based methods (e.g., EoH, ReEvo).
>
> These consistent, SOTA-level results (detailed in Tables 1, 5, 6, and 7) collectively serve as strong empirical evidence that the CoR framework effectively balances exploration and exploitation in practice, leading to reliable convergence toward high-quality solutions.
>
>
> ## Q2 Is there any mechanism (e.g., damping or adaptive scaling) to prevent over-adjustment of hyperparameters?
>
> Indeed, this stability is governed by a pivotal mechanism intrinsic to our framework, orchestrated synergistically by the Actuator LLM and the Experience Pool:
>
> 1. Empirically-Driven Magnitude Determination: As detailed in Section 3.3, a primary function of the Actuator LLM is to determine the precise Magnitude of Adjustment.
>
> 2. In-Context Learning (ICL): The LLM retrieves "analogous historical states" from the experience pool to contextualize its decision. Through ICL, it infers dynamic heuristics, such as "prioritizing small, incremental updates during stable progress" versus "deploying aggressive shifts to escape deep stagnation."
>
> Consequently, the LLM’s reasoning, grounded in historical data, functions as an "adaptive damping" mechanism. Unlike rigid, hard-coded rules (e.g., "never exceed a 5% change"), this approach employs intelligent, real-time scaling, effectively preventing ineffective fluctuations and mitigating the risk of severe parameter oscillation.

---

> ### Author Response · Authors · 2025-11-23
> **Official Comment by Authors(2/2)**
>
> ## W3 Visualization & Interpretability
>
> To demonstrate the interpretability of AutoEP, we visualize the evolutionary trajectory of GA hyperparameters (Mutation and Crossover probabilities) on the TSP-400 instance. The resulting curves reveal a sophisticated, state-aware control strategy that distinctively contrasts with manual, monotonic schedules.
>
> 1. Response to ELA Features (Attention to State): The LLM's decisions are explainable through the observed ELA metrics. Periods of rapid parameter fluctuation correspond to specific landscape characteristics. For example, when the Dispersion Ratio ($D_{ratio}$) indicates a single-funnel structure, AutoEP prioritizes exploitation. Conversely, when Skewness signals a risk of premature convergence, the system triggers a 'rescue' behavior by sharply increasing the mutation rate.
>
> 2. Operationalizing the Exploration-Exploitation Trade-off: The visualization confirms that AutoEP understands the mechanics of GA. The crossover and mutation curves often mirror each other (inverse correlation). This validates that the CoR engine  successfully translates abstract directives (e.g., 'intensify exploration') into logically consistent numerical actions (High Mutation / Low Crossover).
>
> 3. Emergence of Search Policies via In-Context Learning: Beyond direction, the magnitude of adjustments shows structured patterns. The LLM utilizes the Experience Pool to determine not just what to change, but how much. This results in distinct phases: aggressive large-step adjustments to escape stagnation, followed by fine-grained small-step tuning for local convergence, proving the effective utilization of historical search context.
>
>
> The above visual revisions can be found in Section 4.6 of our latest revised paper.
>
> ## Q3 Generalizability to DE/CMA-ES
>
> This accurately reflects the flexibility of AutoEP’s modular design. Extending the framework to a new optimizer, like Differential Evolution (DE), is efficient and requires only a single one-shot prompt. This low overhead is driven by two features:
>
> - Algorithm-Agnostic ELA: Our ELA features (e.g., Skewness, Kurtosis, $R^2$, $D_{ratio}$) describe the state of the population and the fitness landscape rather than the internal mechanics of a specific algorithm. As a result, they can be applied to any population-based metaheuristic without modification.
>
>
> - Pluggable CoR Architecture: The Chain-of-Reasoning (CoR) module is built for extensibility. The Analyst and Actuator LLMs remain unchanged because their tasks (diagnosis and execution) are generic. The only change required is within the Strategist LLM, which needs a one-time setup prompt to define the new algorithm’s hyperparameters (e.g., "Analyze how the scaling factor $F$ and crossover rate $CR$ in DE influence the exploration-exploitation balance").
>
> Our successful deployment across three distinct paradigms (GA, PSO, and ACO) offers empirical evidence of this generalizability and confirms that expanding to new algorithms requires very little manual effort.
>
> ## W4 Baselines: Comparison with EvoLLM
>
> We appreciate the reviewer's suggestion. We recognize EvoLLM as a seminal contribution to continuous optimization, specifically for its novel use of LLMs as an evolution strategy to iteratively refine the mean of a Gaussian distribution.
>
> However, a fundamental distinction exists: our research focuses on discrete combinatorial optimization problems (COPs). EvoLLM's methodology, designed for continuous spaces, is not directly applicable to the discrete domains addressed in this study. Nevertheless, acknowledging its status as a pioneering work in LLM-assisted metaheuristics, we have incorporated a discussion of EvoLLM into the Related Work section to provide a more comprehensive literature review.
>
> ## Q4 Could you report sensitivity analyses for Experience Pool size and CoR decision frequency?
>
>
> The sensitivity analysis for the CoR was previously addressed in Section 4.5 ("Sensitivity to Adjustment Frequency"). In the revised manuscript, we have expanded this section to include a new sensitivity analysis regarding the experience pool length. Please refer to the updated Section 4.5 for the detailed results.
>
> ##  Q5 Latency Breakdown
>
> We apologize for the typographical error in the manuscript; the actual inference latency is 0.3s, not 30ms.
>
> As detailed in Appendix D, our experimental setup utilizes the Qwen3-30B model deployed locally on a server equipped with 4 times A800 GPUs, accessed via a local API to minimize network overhead. The per-decision latency of 0.3s is derived from the following breakdown:
>
> - Strategist LLM: Executed only once per optimization session (initialization phase), thus excluded from the recurring decision latency.
>
> - Analyst LLM: Runs at every decision step with an average inference time of $\approx$ 0.1s.
>
> - Actuator LLM: Runs at every decision step with an average inference time of $\approx$ 0.2s.

---

### Official Review · Reviewer_LhpG · 2025-10-31

**Soundness:** 3
**Presentation:** 3
**Contribution:** 3
**Rating:** 6
**Confidence:** 4

**Summary:**

This paper proposes a LLM-based algorithm configuration framework with reasoning chain. The decision of configurations is decomposed into a multi-step reasoning chain. Starting from a strategist LLM which analyses the impact of the hyperparameters of the given algorithm, LLM analyses the current optimization status from ELA features, determine to enhance exploration or exploitation, and modify the algorithm hyperparameters to achieve that. Experimental results show that the proposed method surpasses RL-based algorthm configuration baselines and LLM-based metaheuristic methods on TSP, CVRP and UAV trajectory optimization tasks.

**Strengths:**

1. The algorithm configuration process is decomposed into several steps with detailed descriptions, which significantly enhances interpretability. Users can deeply analyze why the LLM makes a particular decision and how the framework achieves SOTA performance, providing insights for future algorithm design.

2. The plug-and-play approach is adaptive and generalizable, enabling application across broader optimization scenarios and enhancing the interpretability and performance of LLMs in diverse applications.

3. The experimental results validate the SOTA performance of the proposed method on three combinatorial optimization tasks.

**Weaknesses:**

1. The strategist LLM requires users to provide the name of the target algorithm and its hyperparemeters. It demands users' expert knowledge requirement on the target optimization tasks. For optimization tasks where users lack familiarity or no known promising algorithms exist, specifying a suitable algorithm becomes challenging.

2. Furthermore, the algorithmic knowledge embedded in LLMs is inherently limited and potentially outdated. Consequently, LLMs may struggle to accurately analyze the qualitative effects of hyperparameters in complex, advanced algorithms, which could hinder the framework's generalization across diverse optimization tasks.

**Questions:**

1. The feature description explicitly instructs the LLM to explore or exploit based on whether feature values are high or low. If we already possess this deterministic knowledge about when to increase exploration or exploitation, why not replace this component with a simpler, more efficient machine learning model, such as a decision tree?

2. The feature descriptions and experience pool appear to consume a significant number of tokens. How is the average inference latency per decision maintained within 30 ms despite this potential computational overhead?

3. In Figure 4, AutoEP with different base LLMs demonstrates nearly identical performance. Does this imply that different LLMs are making the same configuration decisions? Why would LLMs of varying scales and capabilities take identical actions?

---

> ### Author Response · Authors · 2025-11-23
> **Official Comment by Authors(1/2)**
>
> We are grateful for the reviewer's insightful feedback and their endorsement of AutoEP's key strengths: interpretability, plug-and-play modularity, and superior performance. We value these constructive suggestions and have provided detailed responses below to clarify our framework's robustness and justify our design choices.
>
> ## W1 User expertise requirement (specifying algorithm/hyperparameters).
>
> We want to clarify that AutoEP is designed specifically as a hyperparameter control framework, rather than a tool for algorithm selection. This distinction stems from three key design considerations:
>
> - Workflow Alignment: In standard optimization pipelines, practitioners usually choose a suitable baseline (such as GA or PSO) before they begin tuning. Therefore, AutoEP focuses on automating the labor-intensive tuning phase that follows, rather than the initial choice of algorithm.
>
> - Lowering the Expertise Barrier: To make the system more accessible to novices, the Strategist LLM uses its pre-trained knowledge to handle the details. Users simply provide the algorithm's name (e.g., GA), and the LLM automatically identifies key hyperparameters and understands their specific roles (for instance, recognizing that mutation rate enhances exploration).
>
> - Expert Flexibility: Conversely, the framework supports manual input, allowing researchers to define custom protocols for novel algorithms. This balance ensures that AutoEP effectively serves both novices, who benefit from automated defaults, and experts, who require customizable configurations.
>
>
> ## W2 LLMs' limited/outdated knowledge on complex algorithms.
>
> We acknowledge the inherent knowledge limitations of LLMs. However, within the specific domain of standard meta-heuristic algorithms (MHAs), LLMs demonstrate robust capability in mapping hyperparameters to exploration and exploitation behaviors. This capability significantly lowers the entry barrier for novice users.
>
> Furthermore, AutoEP is designed with flexibility in mind. For novel algorithms where the LLM might lack prior knowledge, expert researchers can manually define these mappings, effectively bypassing the Strategist LLM. It is crucial to note that the core contribution of AutoEP lies in the real-time, ELA-guided adjustment of parameters, rather than merely their initial characterization.
>
> Regarding the concern that "LLMs may struggle to analyze complex algorithms," we agree that this holds true for "black-box" models. However, MHAs are fundamentally distinct: they are explicitly designed with control parameters that have a direct correspondence to exploration and exploitation. While LLMs might struggle with the opacity of other complex architectures, the intrinsic interpretability of MHA hyperparameters mitigates this concern, rendering them well-suited for LLM-based analysis.
>
> ## Q1 Why not replace LLM with a Decision Tree (or rule-based model) given the feature descriptions?
>
> We highlight three fundamental distinctions that justify our choice of LLMs over Decision Trees (DTs) or RL:
>
> 1. Zero-Shot vs. Supervised Learning: DTs and RL policies require extensive training data. Constructing a universal dataset of labeled "state $\to$ optimal action" pairs is computationally prohibitive and often fails to generalize across diverse algorithms. In contrast, AutoEP operates as a training-free framework, bypassing the need for data collection.
>
> 2. Handling Ambiguity via Fuzzy Reasoning: ELA features frequently present contradictory signals (e.g., high diversity co-occurring with stagnation). Rigid, rule-based systems like DTs struggle to resolve these conflicts without exponentially complex branching. LLMs, however, employ fuzzy reasoning to weigh such trade-offs and infer optimal adjustments by contextualizing historical decision patterns.
>
> 3. Continuous Magnitude Control: As detailed in Section 3.3, the Actuator LLM analyzes historical sensitivity within the experience pool to determine precise adjustment magnitudes (e.g., $+0.01$ vs. $+0.1$). Unlike simple classification trees, which are limited to discrete outputs, AutoEP inherently handles continuous value regression without requiring instance-specific training.

---

> ### Author Response · Authors · 2025-11-23
> **Official Comment by Authors(2/2)**
>
> ## Q2 How is 30ms inference latency maintained with long prompts?
>
> We apologize for the typographical error in the manuscript; the actual inference latency is 0.3s, not 30ms.
>
> As detailed in Appendix D, our experimental setup utilizes the Qwen3-30B model deployed locally on a server equipped with 4 times A800 GPUs, accessed via a local API to minimize network overhead. The per-decision latency of 0.3s is derived from the following breakdown:
>
> - Strategist LLM: Executed only once per optimization session (initialization phase), thus excluded from the recurring decision latency.
>
> - Analyst LLM: Runs at every decision step with an average inference time of $\approx$ 0.1s.
>
> - Actuator LLM: Runs at every decision step with an average inference time of $\approx$ 0.2s.
>
> Consequently, the total per-step latency is the cumulative sum of the serially executed Analyst and Actuator modules ($\approx$ 0.3s).
>
>
> ## Q3 Why do different LLMs (Fig 4) perform almost identically?
>
> The performance consistency across models (Qwen, DeepSeek, GPT-4) highlights the robustness of AutoEP, driven by three factors:
>
> - Reduced Cognitive Load: CoR decomposes the optimization task into simple logical comparisons. By asking the LLM to "follow rules" rather than "solve the problem," we allow smaller models (Qwen3-30B) to match the control capabilities of GPT-4.
>
> - High-Fidelity Input: ELA features provide a precise quantitative basis for decision-making. Since the input signal (e.g., state description) is unambiguous, different models consistently derive the same logical actions (e.g., "stagnation $\to$ mutation").
>
> - Self-Correcting Feedback Loop: The closed-loop mechanism, facilitated by the experience pool, drives convergence toward effective configurations. Even if a model makes a suboptimal decision, immediate negative feedback in subsequent steps triggers correction. This iterative alignment ensures that all models exhibit convergent long-term optimization trajectories.

---

### Official Review · Reviewer_yVzs · 2025-11-01

**Soundness:** 3
**Presentation:** 3
**Contribution:** 3
**Rating:** 8
**Confidence:** 4

**Summary:**

This work describes a novel idea for dynamic algorithm configuration in optimization. The authors are motivated by the limitation of human-crafted configuration rules and generalization bottoleneck of latest learning-assisted approaches. To address these technical issues, the authors propose leveraging LLMs (with vast knowledge) to kick a balance between the configuration effectiveness and generalization. Specifically, the overall framework comprises a ELA calculator and a multi-role LLM system. Given an optimizer and corresponding optimizee at hand (e.g., GA for solving TSP), the ELA calculator summarizes per step dynamic landscape information and feeds such information into the multi-role LLM system. Then the LLM system analyse such information and dictates proper algorithm configuration for the optimizer to control the exploration and exploitation strength. Through extensive experiments, the effectivess of the proposed framework is validated.

**Strengths:**

1. Novelty: While some previous works may also considered using LLM for algorithm configuration or selection such as LLaMoCo (implicit and static algorithm configuration through code generation, https://arxiv.org/pdf/2403.01131) and AS-LLM (training algorithm selection mapping through LLM, https://openreview.net/forum?id=l7aD9VMQUq), I have to acknowledge this paper as a novel perspective, with reasonable usage of LLMs to relieve human efforts required for tuning optimization algorithms. Such novelty roots from the reasoning ability of existing LLMs.

2. Systematic Design: I think the most important and significant contribution of this paper is that it demonstrates that through ELA anaysis, the strength of LLM and the performance of an optimizer can be bridged in a dynamic algorithm configuration fashion.

3. Significance: the experiments results are solid, and especiall compared with latest EoH and ReeVO (LLM-based AAD methods), the performance gain is good.

**Weaknesses:**

Besides the strength of this paper, I wanna list several concerns.

1. Efficiency: I understand that using LLM to provide ideal performance is already a breakthrough for automated algorithm configuration. However, I suggest the authors carefully claim their approach's superiority. Here, an interesting question is since AutoEP has to use LLM for reasoning, why the running time is still comparable with GA itself?

2. The experience pool compreises per step action-reward entries, while it may provides certain heuristic on how to choose parameter value, it is still unclear why such myopia information could help AutoEP bypass RL-based approach such as GLEET, since in RL-based approaches, the bellman updates ensure the policy to learn long-term decision. This deserves in-depth investigation.

3. The underlying optimzers are restricted within simple optimizer such as GA, PSO.... This might makes readers curious about the true potential of AutoEP on more complex optimizer with complex configuration space.

4. The related work could be polished and refined further. Can the authors use Meta-Black-Box Optimization to denote learning-assisted or data-driven or reinforcement learning-based approaches? since recently these methods have been summarized and termed by this terminology. Also consider reviewing latest works in  Meta-Black-Box Optimization (surveys and papers in 2025).

**Questions:**

See Weaknesses.

---

> ### Author Response · Authors · 2025-11-23
>
> We sincerely thank the reviewer for their constructive comments and for recognizing the novelty and systematic design of AutoEP. We are encouraged by the reviewer's endorsement of our core strategy: bridging LLM reasoning with optimizer performance via Exploratory Landscape Analysis. In the following sections, we address specific concerns regarding computational efficiency, comparisons with Reinforcement Learning, and the scope of applicable optimizers.
>
> ## W1 Efficiency: Why is runtime comparable to standard GA? (Efficiency Concern)
>
> We appreciate the reviewer's scrutiny regarding computational cost. We acknowledge that AutoEP incurs overhead due to LLM inference (e.g., 5.8 min vs. 3.3 min baseline on eil51, Table 1). However, our design philosophy prioritizes a high performance-to-cost ratio rather than zero overhead.
>
> The efficiency of AutoEP is structurally guaranteed by our CoR architecture, which enables the use of lightweight open-source models (e.g., Qwen3-30B) instead of computationally heavy proprietary models. Local deployment further eliminates API network latency, reducing per-adjustment inference time to just 0.3 seconds (Appendix D). The impact of this design is evident in the ablation study (Table 3): AutoEP is an order of magnitude faster (5.8 min) than a comparable GPT-4 based implementation (44.7 min), proving that our framework successfully balances reasoning capability with computational feasibility.
>
> ## W2 Mechanism Analysis: Myopic Reasoning vs. RL's Long-Term Policy
>
> We thank the reviewer for this profound insight, which touches upon the fundamental distinction between our approach and traditional RL methods. You are absolutely correct that the experience pool provides "myopic" information, whereas RL seeks to approximate long-term value. AutoEP bypasses the RL training paradigm by substituting the tabula rasa learning process with the pre-trained semantic priors of an LLM.
>
> 1. The RL Paradigm (e.g., GLEET): RL requires learning a policy $\pi(s) \to a$ from scratch, necessitating millions of executions (high sample complexity) to propagate long-term value via Bellman updates. The resulting policy is typically frozen for inference, often suffering from poor generalization and overfitting to the training distribution.
>
> 2. The AutoEP Paradigm: In contrast, AutoEP is a zero-shot, online reasoning framework.
>
> - LLM as Semantic Prior: The LLM has already internalized abstract concepts such as "convergence," "stagnation," and "diversity" during pre-training. It does not need to learn the strategy "increase exploration to avoid stagnation" via iterative updates; it possesses this knowledge a priori.
>
> - Experience Pool as Context: Instead of training a value function, the experience pool facilitates In-Context Learning. As detailed in Section 3.3, the Actuator LLM queries the pool not to learn a policy, but to infer effective adjustment magnitudes from similar historical states. This acts like a domain expert consulting an operation log during a specific run ("In the last low-diversity state, increasing mutation by 0.1 worked well"), rather than an agent training a fixed long-term policy.
>
>
> In summary, while RL strives to learn a policy, AutoEP contextualizes a pre-trained, generalized policy using ELA and the experience pool.
>
> ## W3 Applicability to Complex Optimizers (Scope Concern)
>
> AutoEP is designed with inherent generalizability, allowing seamless extension to advanced optimizers like MOEAs and DE variants. This flexibility is grounded in two principles:
>
> 1. Universal State Representation: By employing optimizer-agnostic ELA features (e.g., $S, K, R^2, D_{ratio}$), we characterize the fitness landscape and population distribution rather than specific algorithmic mechanics, making the framework compatible with any population-based metaheuristic.
>
> 2. Modular CoR: Our architecture adopts a "plug-and-play" design:
>
> - The Strategist adapts to new optimizers via one-shot prompting to learn hyperparameter semantics (e.g., the effects of $F$ and $CR$ in DE).
>
> - The Analyst remains algorithm-independent, focusing strictly on state diagnosis.
>
> - The Actuator executes adjustments based on the Strategist’s updated mapping. This modularity is central to AutoEP, facilitating scalable deployment across varying optimization paradigms.
>
>
> ## W4 Related Work & Terminology (Polishing)
>
> We appreciate the reviewer's professional suggestion. Accordingly, we have renamed the relevant subsection in the Related Work to "Meta-Black-Box Optimization" and have incorporated a broader review of the pertinent literature. These updates are reflected in the revised manuscript.

---

> > ### Comment · Reviewer_yVzs · 2025-11-24
> > **further comment**
> >
> > My further comments focus on W2 and W3.
> >
> > W2: I agree with the authors that LLM may has prior on the algorithm performance description, which hence supports in-context algorithm configuration. However, I also noted that RL-based approach such as GLEET is tailored for continuous optimization problem, hence its design may not reflects problem landscape of COPs. I think the scenario the authors chose (COP with time-consuming running) makes AutoEP reflects its value, that is, in a time-consuming scenario, the time budget LLM uses to configure per step hyper-parameters can be neglected, but the direct, in--context, maybe myopic policy extraction could already provide an acceptable performance boost. I think this aspect deserves a small discussion in your paper.
> >
> > W3: I understand the universal workflow of AutoEP. What I am expressing is that in practice, there are many novel optimizers which were not in the LLM's prior. To understand and reasoning the hyper-parameters in such optimizer, LLM would require additional knowledge (maybe from an expert's description). How could AutoEP ensure the robust performance under such cases? Note: I am not challenging this work's novelty, I expect a discussion with the authors, and prompt accordingly some future directions would be better.

---

> > > ### Author Response · Authors · 2025-11-25
> > >
> > > We appreciate the opportunity to further discuss these points. The reviewer’s insights regarding computational trade-offs (W2) and the handling of novel optimizers (W3) are astute and accurately capture the core design philosophy of AutoEP. We look forward to integrating these discussions into the final manuscript.
> > >
> > >
> > > ## Response to W2
> > >
> > > We fully concur with the reviewer's assessment. The utility of AutoEP is indeed maximized in scenarios where objective function evaluations are computationally expensive (e.g., large-scale COPs or complex simulations like the UAV trajectory case). In such high-latency regimes, the inference cost of the LLM (milliseconds to seconds) becomes negligible relative to the fitness evaluation time (minutes to hours). We have incorporated this critical distinction into the Discussion section.
> > >
> > >
> > >
> > > ## Response to W3
> > >
> > > While new meta-heuristics introduce hyperparameters that may be unknown to the Strategist LLM, AutoEP addresses this issue through a decoupled architecture that facilitates expert integration.
> > >
> > > The design isolates the algorithm-specific logic: the Strategist is the sole module responsible for mapping hyperparameters to search behaviors (exploration vs. exploitation). Conversely, the Analyst operates on algorithm-agnostic ELA features, and the Actuator simply executes adjustments based on the Strategist's guidance. Consequently, integrating a novel algorithm requires only one manual step: a domain expert must configure the Strategist to establish this mapping. Once defined, the Analyst and Actuator function automatically, maintaining the framework's stability.

---

> > > > ### Comment · Reviewer_yVzs · 2025-11-25
> > > > **final comments**
> > > >
> > > > Much appreciate for the responses. I will keep my score and recommend this paper for acceptance.

---

> > > > > ### Author Response · Authors · 2025-11-25
> > > > >
> > > > > We sincerely thank the reviewer for the positive assessment and the recommendation for acceptance. We are grateful for your time and the constructive feedback provided throughout the discussion, which has been invaluable in strengthening our work.

---

### Author Response · Authors · 2025-12-01
**Summary**

To the Area Chair,

We thank the reviewers for their constructive feedback and their unanimous support for our work (Scores: 8, 6, 6, 6). The reviews collectively recognize AutoEP as a novel, methodologically sound, and effective framework that bridges Large Language Models (LLMs) with Exploratory Landscape Analysis (ELA) for zero-shot algorithm control.

We provide a summary of the key points raised during the review process and how they support the acceptance of this paper:

1. Novelty and Significance (Consensus Strength).  All reviewers (yVzs, LhpG, yTaP, xA6g) acknowledged the novelty of shifting from "learning-to-optimize" (RL-based) to "reasoning-to-optimize" (LLM-based). Reviewer yVzs highlighted the systematic design of bridging LLM reasoning with optimizer performance via ELA as a significant contribution. The extensive experiments demonstrating SOTA performance on TSP, CVRP, FSSP, and UAV trajectory optimization were praised for their rigor.

2.	Robustness and Reproducibility. We demonstrated that AutoEP is model-agnostic, achieving SOTA performance with smaller open-source model (Qwen3-30B) where other methods (EoH, ReEvo) fail without GPT-4. This ensures the framework is accessible and reproducible for the wider research community.

3.	We have successfully resolved all technical issues during the rebuttal. Reviewer yVzs now recommends acceptance, and Reviewer xA6g retains a positive score. Regarding Reviewers LhpG and yTaP, who have not yet replied: they initially recognized the work as novel and generalizable. We have thoroughly addressed their specific questions on scalability, efficiency, and visualization, integrating these improvements into the revision.

Detailed Response Summary:

**Computational Efficiency and Latency (Reviewers yVzs, LhpG, yTaP):**

Concern: Queries regarding the inference cost and a typographical error in the manuscript stating "30ms" latency.

Resolution: We clarified that the actual per-step latency is 0.3s (not 30ms). We emphasized that AutoEP is designed for local deployment using efficient open-source models (e.g., Qwen3-30B), making it orders of magnitude faster than proprietary API-based solutions (5.8 min vs. 44.7 min for GPT-4). The reviewers accepted that this latency is negligible for computationally expensive optimization tasks (e.g., UAV trajectory planning).

**Reasoning vs. Reinforcement Learning (Reviewer yVzs):**

Concern: How AutoEP’s "myopic" experience pool compares to RL’s long-term policy learning.

Resolution: We clarified the paradigm shift: AutoEP bypasses the high sample complexity of training RL policies from scratch by leveraging the semantic priors of pre-trained LLMs. The experience pool facilitates In-Context Learning, allowing the model to adapt dynamically without freezing a policy. Reviewer yVzs agreed this is a valuable approach for scenarios where training overhead is prohibitive.

**Memory and Context Management (Reviewer yTaP):**

Concern: Potential context bloat in the Experience Pool during long runs.


Resolution: We clarified our implementation of a sliding window mechanism ($N=20$) and provided new sensitivity analyses showing this optimal trade-off between trajectory awareness and context noise/latency.


**Generalization and Complex Scenarios (Reviewer xA6g, yTaP):**

Concern: Whether the method works on "new" tasks or complex constraints beyond standard benchmarks.

Resolution: We directed attention to our UAV Trajectory Optimization experiments (a complex, real-world problem with dynamic constraints), where AutoEP achieved a 17.16% performance gain over baselines. We also explained how the "Strategist" module allows for one-shot adaptation to new algorithm families (e.g., DE) without retraining.

**Conclusion**
AutoEP presents a plug-and-play, training-free paradigm that significantly advances Automated Algorithm Design (AAD). Given the positive consensus, the resolution of technical queries, and the demonstrated superior performance over both RL and existing LLM-based methods, we believe this work is well-positioned for inclusion in ICLR.

Best regards,

The Authors

---

### Meta-Review · Area_Chair_3ANU · 2025-12-29

**Summary:**

The reviewers unanimously concluded that the manuscript is suitable for publication.

**Reviewer Concerns:**

no concerns

**Reviewer Scores:**

not relevant

---

### Decision · Program_Chairs · 2026-01-26

Accept (Oral)